# REG: In-Sample RL via Regularizing the Evaluation Gap

**Hanpu Shen** [1]   **Weining Shen** [1]   **Roy Fox** [2]

## Abstract

Distribution shift poses a fundamental challenge in offline reinforcement learning, often leading to value overestimation when querying out-of-distribution actions. We introduce Regularized Evaluation Gap (REG) as a bridge between implicit methods like IQL and explicit conservative methods. We formulate policy evaluation as a robust optimization problem over an ambiguity set of critics and show that IQL's objective can be viewed as an approximate dual solution to this problem. To extract a policy from the learned value function, we propose a practical Orthogonal Policy Gradient (OPG) update. This method regularizes an aggressive, mode-seeking policy gradient by projecting it onto the subspace orthogonal to a stable, in-sample behavior cloning gradient. Extensive D4RL experiments demonstrate that REG matches state-of-the-art performance among both Gaussian methods and diffusion-based approaches without the computational burden of the latter.[1]

## 1. Introduction

Reinforcement learning (RL) is a cornerstone technology for developing highly capable AI systems. It has achieved significant success in challenging game-playing domains (Mnih et al., 2013; Silver et al., 2017) and in fine-tuning Large Language Models for complex downstream tasks (Wang et al., 2024; Xu et al., 2024; Guo et al., 2025). However, the online-learning paradigm is often limited by the high cost and risk of interactions with real-world environments and feedback providers. Offline RL addresses these challenges by focusing on training agents solely from fixed datasets of environment interactions and their rewards. By removing the requirement for online exploration, offline RL can be applied to a wide range of critical domains, such as autonomous driving (Kiran et al., 2021; Li et al., 2024) and healthcare (Tang & Wiens, 2021; Zhou, 2023; Shen et al., 2026), where large amounts of previously logged data are available.

A fundamental challenge in offline RL is the distribution shift between the learned policy and the behavior policy that generated the dataset (Fujimoto et al., 2018). Directly applying off-policy algorithms by filling replay buffers with offline data fails because the policy improvement step queries the learned Q-function on out-of-distribution (OOD) actions. Since the Q-function is untrained in these states, its estimates are unreliable and prone to optimistic extrapolation errors, leading to poor deployed performance (Kumar et al., 2020). To mitigate this, model-free offline RL algorithms primarily adopt the principle of OOD pessimism. This is typically achieved through explicit conservativeness, which regularizes the policy to remain close to the data distribution (Nachum et al., 2019; Wang et al., 2023; Tarasov et al., 2023; Liu et al., 2024) or introduces regularization to the value-estimation objective (Kumar et al., 2020; Kostrikov et al., 2021a; Luo et al., 2025; Zhang et al., 2024; Shimizu et al., 2024). Alternatively, recent studies have adopted implicit conservativeness to avoid OOD action queries altogether (Kostrikov et al., 2021b; Hansen-Estruch et al., 2023; He et al., 2024; Xu et al., 2023).

Implicit Q-Learning (IQL) (Kostrikov et al., 2021b) has emerged as a particularly successful implicit paradigm. It avoids OOD queries by training a state-value function $V(s)$ to approximate the maximum Q-value over in-sample actions, then uses $V(s')$ as a stable, in-sample Bellman target for the Q-function. This is achieved via expectile regression—an effective but ultimately heuristic choice. Its successor, IDQL (Hansen-Estruch et al., 2023), achieved state-of-the-art performance by maintaining this implicit paradigm while leveraging expressive diffusion policies instead of standard Gaussian ones trained with Advantage-Weighted Regression (AWR) (Peng et al., 2019). Other frameworks, such as Implicit Value Regularization (IVR) (Xu et al., 2023) and various DICE-based methods (Nachum et al., 2019; Lee et al., 2021; Mao et al., 2024b), utilize policy regularization formulations that introduce divergence terms to constrain the policy to the behavior distribution.

[1]Department of Statistics, University of California Irvine, Irvine, United States [2]Department of Computer Science, University of California Irvine, Irvine, United States. Correspondence to: Hanpu Shen <hanpus1@uci.edu>, Roy Fox <royf@uci.edu>.

*Proceedings of the 43$^{rd}$ International Conference on Machine Learning*, Seoul, South Korea. PMLR 306, 2026. Copyright 2026 by the author(s).

[1]Our code is available at `https://github.com/HanpuShen/reg_offline_rl`.

This motivates a more direct line of inquiry: instead of regularizing the policy, can we derive in-sample learning by directly addressing the root cause of the problem—unreliable out-of-distribution value estimation? If critic overestimation is the core issue, a more fundamental approach is to constrain the properties of the value function itself. We seek a policy whose performance is provably bounded by a set of plausible critics, constrained to be consistent with the data.

We introduce the Regularized Evaluation Gap (REG) method, which reframes in-sample learning as a principled robust optimization problem. The key idea is to find a policy whose value can be reliably estimated across an ambiguity set of plausible critics that are consistent with the offline data. By applying convex duality to this constrained problem, we derive a tractable learning objective. Notably, our derivation reveals that an IQL-like one-sided regression objective is not merely a heuristic, but can be viewed as an approximate dual solution to this robust formulation. Our main contributions:

1. We demonstrate that the solution of REG yields a one-sided regression objective for value learning. This provides a principled theoretical foundation for the asymmetric updates used in IQL and offers a new perspective on why such updates are effective.

2. We propose a practical algorithm based on REG, featuring a novel Orthogonal Policy Gradient (OPG) update. This method adaptively balances an in-sample weighted behavior cloning gradient with a mode-seeking probing policy gradient.

3. We provide extensive empirical evidence showing that our algorithm matches the state-of-the-art performance among Gaussian-policy methods on the D4RL benchmark suite. It is also competitive with more complex diffusion-based approaches while utilizing a significantly simpler and more efficient architecture.

## 2. Related Works

Offline reinforcement learning algorithms are predominantly designed around the principle of pessimism to counteract value overestimation for out-of-distribution (OOD) actions (Levine et al., 2020). This pessimism can be realized through several distinct algorithm families.

**Explicit Conservativeness.** One major line of work introduces explicit regularization to enforce conservative value estimates (Fujimoto & Gu, 2021; Kumar et al., 2020; Tarasov et al., 2023; Wang et al., 2023; 2020; Lyu et al., 2022). Conservative Q-Learning (CQL) (Kumar et al., 2020) directly penalizes high Q-values for OOD actions, though this can require computationally intensive sampling to estimate the regularizer. Another approach constrains the learned policy to stay close to the behavior policy. This is often achieved

by adding a behavioral cloning (BC) term, as seen in simple yet effective methods like TD3+BC (Fujimoto & Gu, 2021), or by constraining a divergence metric between the policies, as in ReBRAC (Tarasov et al., 2023). Recent works such as Diffusion Q-learning (Wang et al., 2023) use a diffusion model to parametrize the actor in a TD3-BC style algorithm.

**Distribution Matching.** Pioneered by distribution-correct estimation (DICE) methods (Nachum et al., 2019; Lee et al., 2021; Mao et al., 2024a;b; Liu & Hofert, 2024), this approach seeks to directly optimize for the discounted state-action occupancy of the learned policy. By framing this as a constrained optimization problem solved via a convex dual, methods like OptiDICE (Lee et al., 2021) can learn value functions without bootstrapping from a learned critic, thereby remaining in-sample. ODICE (Mao et al., 2024b) proposes a novel value optimization paradigm for DICE that leverages the forward and backward gradient and achieves state-of-the-art performance among Gaussian-policy methods. More recent works leverage generative models to represent the behavior policy support, such as Diffusion-DICE (Mao et al., 2024a), though these methods can be computationally demanding at both training and deployment time.

**Implicit Conservativeness.** In contrast to explicit regularization, implicit methods avoid OOD queries altogether by using asymmetric loss functions for the Bellman update (Kostrikov et al., 2021b; Garg et al., 2023; Hansen-Estruch et al., 2023; Xu et al., 2023; 2025). This paradigm was popularized by Implicit Q-Learning (IQL) (Kostrikov et al., 2021a), which uses expectile regression to learn a state-value function that implicitly targets the maximum Q-value over in-sample actions. This idea has since been generalized; for example, Extreme Q-Learning (EQL) (Garg et al., 2023) provides a statistical interpretation based on the Gumbel distribution, and Implicit Value Regularization (IVR) (Xu et al., 2023) connects IQL-style updates to behavior-regularized MDPs. Implicit Diffusion Q-learning (IDQL) (Hansen-Estruch et al., 2023) implements IQL policy extraction on a diffusion actor to achieve state-of-the-art results across offline RL benchmarks.

Our work, Regularized Evaluation Gap (REG), offers a bridge between implicit and explicit conservativeness via a principled optimization problem. Similar to DICE and IVR, we employ convex duality to solve a constrained problem. Unlike DICE, which constrains occupancy measures, and IVR, which instead regularizes the policy, REG directly constrains the Bellman evaluation error of the critic itself. This formulation as a robust optimization problem on the critic's properties provides a novel theoretical path. Our framework's dual solution yields a one-sided regression objective for value learning, providing a principled alternative to IQL's expectile regression. More importantly, our approach unifies the value and policy updates: the dual solu-

tion not only specifies the value objective but also yields an explicit form of the optimal policy and motivates our novel Orthogonal Policy Gradient (OPG) algorithm that proved instrumental for our method's state-of-the-art performance.

## 3. Preliminaries

We model reinforcement learning (RL) as a discounted Markov decision process (MDP) (Sutton et al., 1998) defined by the tuple $\mathcal{M} = (\mathcal{S}, \mathcal{A}, P, r, \gamma, d_0)$. Here, $\mathcal{S}$ and $\mathcal{A}$ denote the state and action spaces, $P(\cdot \mid s, a)$ is the dynamics, $r(s, a)$ is the reward function, $\gamma \in [0, 1)$ is the discount factor, and $d_0$ is the initial state distribution. A stochastic policy $\pi(a \mid s)$ induces a trajectory distribution $\xi \sim P_\pi$. The objective is to maximize the expected discounted return $J_\pi = \mathbb{E}_{\xi \sim P_\pi}[\sum_{t=0}^{\infty} \gamma^t r(s_t, a_t)]$. This can be expressed as $J_\pi = \mathbb{E}_{(s,a) \sim d_\pi}[r(s, a)]$, where $d_\pi$ is the discounted state–action occupancy measure. In the offline setting, the agent learns from a fixed dataset $\mathcal{D} = \{(s_i, a_i, r_i, s_i')\}_{i=1}^N$ with $r_i = r(s_i, a_i)$ and $(s_i' \mid s_i, a_i) \sim P$, that is collected by a behavior policy $\mu(a \mid s)$, without further interaction with the environment.

**In-sample Implicit Value Learning.** To address the distribution-shift issue in offline RL, in-sample learning avoids querying out-of-distribution (OOD) actions. Ideally, we seek the maximum value over actions within the support of the behavior policy. Let $Q_\theta$ be a critic and $V_\psi$ a state-value network. Implicit Q-Learning (IQL) (Kostrikov et al., 2021a) employs a one-side regression to recover the optimal value function under the dataset support constraints (Kostrikov et al., 2021b, Theorem 3):

$$\max_{a' \text{s.t.} \mu(a'|s')>0} Q_{\bar\theta}(s', a') \tag{1}$$
$$= \arg\min_{V_\psi} \mathbb{E}_{(s,a) \sim \mathcal{D}}\left[\max(0, Q_{\bar\theta}(s, a) - V_\psi(s))^2\right].$$

However, directly optimizing this objective is unstable and prone to overestimation due to aleatoric uncertainty in the dataset (i.e., "lucky" samples). To mitigate this, IQL approximates the in-sample maximization via expectile regression, by minimizing the asymmetric loss $L_2^\tau(u) = |\tau - \mathbb{I}(u < 0)|u^2$ with expectile $\tau \in (0.5, 1)$:

$$\mathcal{L}_V(\psi) = \mathbb{E}_{(s,a) \sim \mathcal{D}}\left[L_2^\tau(Q_{\bar\theta}(s, a) - V_\psi(s))\right]. \tag{2}$$

In the limit $\tau \to 1$, $L_2^\tau$ converges to the right term in (1), connecting expectile regression to in-sample max updates, while still avoiding maximization over out-of-dataset actions.

For policy extraction, IQL uses AWR (Peng et al., 2019) to train a policy network via weighted regression, maximizing

$$\mathcal{L}_{\text{IQL}}(\phi) = \mathbb{E}_{\mathcal{D}}\left[\exp(\beta A(s, a)) \log \pi_\phi(a \mid s)\right], \quad \text{(IQL)}$$

with the advantage $A(s, a) = Q_{\bar\theta}(s, a) - V_\psi(s)$. Similarly motivated, IDQL (Hansen-Estruch et al., 2023) explicitly

constructs the policy implied by the implicit value function minimizing (2)

$$\pi_{\text{IDQL}}(a \mid s) \propto \mu(a \mid s) \cdot \left|\tau - \mathbb{I}(A(s, a) < 0)\right|. \quad \text{(IDQL)}$$

Despite their success, these approaches have limitations. IDQL relies on a direct re-weighting of the behavior policy, necessitating accurate estimation of $\mu(a \mid s)$, which is challenging in high dimensions. Furthermore, these methods extract the *implicit* policy implied by the value function, rather than directly solving for the optimal policy. In contrast to relying on asymmetric losses or separate, potentially misaligned policy extraction steps, we seek a framework that simultaneously derives conservative value updates and an optimal policy extraction rule.

## 4. Offline RL via Regularized Evaluation Gap

In this section, we develop our framework, Regularized Evaluation Gap (REG). We begin by using the standard policy evaluation lemma to pinpoint the fundamental obstacle in offline RL: an inestimable error term arising from distribution shift. Motivated by this, we propose a robust optimization problem that directly constrains this evaluation uncertainty. We then derive the dual of this problem, which yields a tractable, in-sample learning objective and an explicit form for the optimal policy. Finally, we show how instantiating this framework with square loss naturally connects to IQL-style algorithms.

### 4.1. The Challenge of Off-Policy Evaluation

The performance of any policy $\pi$ can be related to an arbitrary critic $q$ through the policy evaluation lemma. This identity is key to understanding the challenge of offline RL.

**Lemma 4.1** (Policy Evaluation). *For any policy $\pi$ and any function $q : \mathcal{S} \times \mathcal{A} \to \mathbb{R}$, the true policy value $J_\pi$ satisfies*

$$J_\pi - \mathbb{E}_{(s,a) \sim d_0, \pi}[q(s, a)] = \tfrac{1}{1-\gamma}\mathbb{E}_{(s,a) \sim d_\pi}[\Delta_\pi q(s, a)]. \tag{3}$$

$\Delta_\pi q(s, a) = r(s, a) + \gamma\mathbb{E}_{(s',a'|s,a) \sim P, \pi}[q(s', a')] - q(s, a)$ *is the Bellman error for policy $\pi$. See Appendix A.2 for a standard proof.*

The lemma shows that, when the expected Bellman error is small, $q$ has the information needed to evaluate the policy well. In an offline setting, we can only access the $d_\pi$-expected Bellman error using data from $d_\mu$. By splitting the expectation on the right-hand of (3) over the support of the behavior data and the region outside it, we get:

$$\underbrace{\mathbb{E}_{d_\mu}\left[\tfrac{d_\pi(s,a)}{d_\mu(s,a)}\Delta_\pi q(s, a)\right]}_{\text{In-Distribution Error}} + \underbrace{\mathbb{E}_{d_\pi}\left[\mathbb{I}(d_\mu(s, a) = 0)\Delta_\pi q(s, a)\right]}_{\text{OOD Error}}.$$

The OOD error term is inestimable from an offline dataset, as it involves expectations over state–action pairs never seen

in the data. The in-distribution error depends on the quality of the critic $q$ and the density ratio, making evaluation highly uncertain. This directly motivates a robust approach: we seek a policy that performs well even under the worst-plausible critic that is consistent with the offline data.

## 4.2. Evaluation-Robust Optimization

Based on the insight from Lemma 4.1, we frame offline RL as a robust policy optimization problem. The goal is to find a policy $\pi$ that maximizes rewards, subject to the constraint that its performance can be reliably evaluated. This is achieved by ensuring the policy's on-policy expected Bellman error remains small, even under the worst-case critic in a set of plausible, data-consistent candidates.

Concretely, we impose a two-level robust constraint on the *learnable* Q-function that measures the Bellman error. This implicitly restricts the policy class to those that are robustly evaluated by any Q-function that is learnable from the offline dataset. Intuitively, we take the first constraint to bound the worst-case evaluation gap of the policy $\pi$, and the second constraint to limit the class of $q$ functions to those that achieve low $\pi$-evaluation loss under the offline distribution, with some convex loss function $f$ that achieves its unique minimum at $f(0) = 0$ (e.g. square loss). The objective is

$$\max_{\pi \in \Pi} \quad \mathbb{E}_{(s,a) \sim d_\pi}[r(s,a)] \tag{4}$$

$$\text{s.t.} \quad \max_{q \in Q(\pi)} \left| \mathbb{E}_{(s,a) \sim d_\pi}[\Delta_\pi q(s,a)] \right| \leq \epsilon_1, \tag{5}$$

$$Q(\pi) = \left\{ q : \mathbb{E}_{(s,a) \sim d_\mu}[f(\Delta_\pi q(s,a))] \leq \epsilon_2 \right\}. \tag{6}$$

Here, $\Delta_\pi q(s, a)$ is the same Bellman error of Lemma 4.1, using the true MDP dynamics (unavailable to the practical algorithm in Section 4.4). Intuitively, $\epsilon_1$ bounds the error from trusting any critic $q \in Q(\pi)$ to evaluate the policy, while $\epsilon_2$ defines how strictly we require the critic to fit the offline data. The set $Q(\pi)$ defines an **ambiguity set** of plausible critics for policy $\pi$. Constraint (5) then ensures that the policy is robustly evaluated, while the ambiguity set in (6) contains all critic functions $q$ that are consistent with the Bellman equation, as measured by the loss $f$ on the offline distribution $d_\mu$.

The interaction between $\epsilon_1$ and $\epsilon_2$ provides a formal mechanism for in-sample learning that governs conservativeness and robustness. Outside the support of the behavior distribution $d_\mu$, the critic $q$ is unconstrained by (6). Consequently, the inner maximization in (5) can choose an arbitrarily large Bellman error for any policy that deviates from the data support, effectively rendering OOD actions infeasible for any finite $\epsilon_1$. Theoretically, setting $\epsilon_2 = 0$ recovers a critic $q$ that is perfectly consistent with the Bellman equation in-distribution but arbitrary elsewhere. However, in practice, the epistemic uncertainty inherent in finite datasets makes such exactness impossible. Once we replace true

expectations with empirical ones, we must allow $\epsilon_2 > 0$ to encompass the estimation error of the critic. This softens the boundary of the in-distribution region, allowing members of the ambiguity set $Q(\pi)$ to disagree within its support but not arbitrarily so. Thus, $\epsilon_2$ balances between overfitting the offline dataset and underfitting it, creating in-distribution evaluation ambiguity where samples are scarce.

Given this ambiguity, we must then correspondingly set a sufficiently large $\epsilon_1 > 0$, to allow the policy to visit state–action pairs where the feasible critics disagree; (5) is unsatisfiable with $\epsilon_2 > 0$ and $\epsilon_1 = 0$. Thus, $\epsilon_1$ controls how conservatively the policy avoids evaluation ambiguity, preventing the policy from exploiting a spuriously optimistic critic that happens to fit the offline data but generalizes poorly. This two-level formulation yields a robust, data-efficient surrogate for policy optimization that can mitigate the overestimation that plagues offline RL in limited-coverage settings.

The primal problem in (4) is intractable to solve directly, as it involves an optimization over policies and their unknown occupancy measures $d_\pi$. We leverage Lagrangian duality to transform it into a tractable, equivalent saddle-point problem that can be solved using only samples from the dataset $\mathcal{D}$. This allows us to optimize over a simpler set of dual variables, including a value function $V$, rather than policies directly.

**Theorem 4.2** (Dual Formulation and Optimal Policy). *Let* $f : \mathbb{R} \to \mathbb{R}_+$ *be a differentiable convex function with minimum* $f(0) = 0$*, that has a "skew direction"* $\kappa \in \{1, -1\}$ *satisfying* $f(\kappa x) \leq f(-\kappa x)$ *for all* $x > 0$*. Then the following optimization problem is equivalent to (4):*

$$\min_{\alpha_1 \geq 0} \max_{\alpha_2 \geq 0} \min_V \left( \mathbb{E}_{d_\mu}\left[ \alpha_1 \alpha_2 \mathbb{I}(A(s,a) \geq 0) f\left( \frac{\kappa A(s,a)}{\alpha_1} \right) \right] \right.$$

$$\left. + (1 - \gamma) \mathbb{E}_{d_0}[V(s)] + \alpha_1 \epsilon_1 - \alpha_1 \alpha_2 \epsilon_2 \right), \tag{7}$$

*where* $A(s,a) = r(s,a) + \gamma \mathbb{E}_{(s'|s,a) \sim p}[V(s')] - V(s)$ *is the advantage. The optimal weighting function, implying the optimal policy* $\pi^*$ *for (4) via* $\omega^*(s,a) = \frac{d_{\pi^*}(s,a)}{d_\mu(s,a)}$*, is given by*

$$\omega^*(s,a) = \max\left( 0, \alpha_2 \kappa f'\left( \frac{\kappa A(s,a)}{\alpha_1} \right) \right),$$

*with* $f$*'s derivative* $f'$*. See A.1 for proof.*

The dual objective is amenable to optimization via standard stochastic gradient methods, as it only requires expectations over the known data distribution $d_\mu$ and the initial state distribution $d_0$. We can view the multipliers $\alpha_1, \alpha_2$ as regularization hyperparameters, to obtain a practical algorithm.

## 4.3. Theoretical Analysis

We now provide a finite-sample analysis of the REG objective. Rather than seeking a tight minimax-optimal bound,

the primary goal of this analysis is to conceptually decompose the suboptimality gap to understand how the primal constraints $(\epsilon_1, \epsilon_2)$ govern the behavior of the learned policy. By studying this decomposition, we can characterize the fundamental trade-offs between conservativeness, robustness, uncertainty, and statistical error inherent in our framework.

**Assumption 4.3.** The reward satisfies $r(s, a) \in [0, R]$ and the state $s \in \mathbb{R}^d$ is normalized to satisfy $\|s\|_2 \leq 1$.

**Assumption 4.4.** The loss function $f$ is convex, $L$-smooth, and achieves a minimum at $f(0) = 0$.

**Assumption 4.5.** The value function $V$ is a member of a function class $\mathcal{F}_V$ with finite Rademacher complexity and is bounded on the normalized state space $\sup_{\|s\|_2 \leq 1} V(s) \leq B$.

Under these assumptions, we present our main theoretical result, a PAC regret bounding the difference between the value of $\pi^*$, the optimal policy supported by $\mu$, and of the policy $\hat{\pi}$ returned by solving the objective (4).

**Theorem 4.6** (Informal). *Set $\epsilon_1$ and $\epsilon_2$ that admit the optimal in-distribution policy $\pi^*$ as feasible for (4), and let $\hat{\pi}$ be the policy obtained by solving the dual objective in Theorem 4.2 using an offline dataset $\mathcal{D}$ of size $n$. Under regularity conditions, with probability at least $1 - \delta$, the regret is bounded by:*

$$J_{\pi^*} - J_{\hat{\pi}} \leq \frac{1}{1 - \gamma} O\left( \underbrace{\epsilon_1 + \alpha_2 \epsilon_2}_{\text{Specification Error}} \right.$$
$$\left. + \underbrace{\alpha_2 \cdot poly(R, B, L)\left( \mathcal{R}_n(\mathcal{F}_V) + \sqrt{\frac{\log(1/\delta)}{n}} \right)}_{\text{Statistical Error}} \right).$$

Theorem 4.6 (see A.3 for details) decomposes the optimality gap into two intuitive components:

- **Specification Error:** The term $\epsilon_1 + \alpha_2 \epsilon_2$ represents the error induced by the problem specification itself. The constraint tolerances $\epsilon_1$ and $\epsilon_2$ define the degree of allowable slack in the primal problem. This term quantifies the bias introduced by relaxing the ideal, unconstrained optimization objective.

- **Statistical Error:** The second term characterizes the estimation error from learning with a finite dataset of size $n$. It depends on the Rademacher complexity of the value function class, $\mathcal{R}_n(\mathcal{F}_V)$, which vanishes as $n \to \infty$. This term captures the variance inherent in the learning process.

To better understand the interplay between these terms, we specialize the bound for the square loss, $f(x) = x^2/2$, and derive a simpler expression (see A.3 for details).

**Corollary 4.7.** *For $f(x) = x^2/2$, the optimality bound is:*

$$J_{\pi^*} - J_{\hat{\pi}} \leq \frac{1}{1 - \gamma} O\left( \epsilon_1 + \sqrt{\epsilon_2} \right.$$
$$\left. + \frac{poly(R, B)}{\sqrt{\epsilon_2}}\left( \mathcal{R}_n(\mathcal{F}_V) + \sqrt{\frac{\log(1/\delta)}{n}} \right) \right).$$

**The Role of $\epsilon_2$ as a Fitness Controller.** The form of the bound in Corollary 4.7 reveals a fundamental bias–variance trade-off governed by $\epsilon_2$. The choice of this constraint tolerance is not merely a specification detail but acts as a critical hyperparameter controlling the solution's sensitivity to statistical uncertainty. A small $\epsilon_2$ reflects high confidence in the critic's fit to the true Bellman equation on the dataset. This tightens the constraint, decreasing the bias term ($\sqrt{\epsilon_2}$) but reduces the effective regularization, which amplifies the statistical error ($1/\sqrt{\epsilon_2}$). Under-regularizing by setting $\epsilon_2$ too low risks overfitting to sampling noise in the dataset. As we increase $\epsilon_2$, on the other hand, we introduce bias by shrinking the set of policies that are conservative with respect to the expanding ambiguity set.

**The Role of $\epsilon_1$ as a Conservativeness Controller.** Here, the finite-sample analysis of Theorem 4.6 and Corollary 4.7 shows real strength. In objective (4), it appears that the optimum is monotonically non-decreasing in $\epsilon_1$, because relaxing constraint (5) admits more policies. This is the correct conclusion in an objective that evaluates policies on the true MDP, but the finite-sample analysis reveals that the opposite is true in practice. Intuitively, with $\epsilon_2 > 0$ forming an in-distribution ambiguity set that regularizes the value where samples are scarce, selecting a large $\epsilon_1$ only serves to ignore that ambiguity rather than be conservative for it. Theorem 4.6 suggests that we should set $\epsilon_1$ as low as possible to decrease the evaluation bias, but not so low that it renders $\pi^*$ infeasible. At the limit of a large dataset, $n \to \infty$, we should set $\epsilon_2 = 0$ to recover the true in-distribution value and then objective (4) is indifferent to any finite $\epsilon_1$. The available theory does not suggest a method for setting $\epsilon_1$ and $\epsilon_2$ to minimize the regret, nor a tight analysis of alternative settings, but empirical studies in Section 5 suggest mild sensitivity to these hyperparameters.

### 4.4. Practical Algorithm

Theorem 4.2 presents a general solution to our robust optimization problem. To translate this into a practical algorithm, we must make several well-motivated modeling choices and approximations. We begin by instantiating the framework with the square loss $f(x) = x^2/2$. This choice corresponds to a Gaussian assumption on the Bellman errors and is common in practice. Substituting $f'(x) = x$ into (7), the

value-function objective becomes

$$\min_V \mathbb{E}_{d_\mu}\left[\frac{1}{2}\max\left(0, r(s,a) + \gamma\mathbb{E}_{(s'|s,a)\sim P}[V(s')] - V(s)\right)^2\right]$$
$$+ \alpha\mathbb{E}_{d_0}[V(s)], \tag{8}$$

with $\alpha = \frac{\alpha_1}{\alpha_2}(1-\gamma)$, and the optimal weighting function

$$\omega^*(s,a) \propto \max(0, r(s,a) + \gamma\mathbb{E}_{s'}[V(s')] - V(s)). \tag{9}$$

This objective provides a principled theoretical justification for the asymmetric updates used in Implicit Q-Learning (IQL) (Kostrikov et al., 2021b). Recall that IQL approximates the in-sample maximum using expectile regression (2). Our derived term $\max(0, u)^2$ is mathematically equivalent to $\lim_{\tau\to 1}\mathcal{L}_2^\tau$. In Appendix B, we validate the connection further by showing that the dual hyperparameter $\alpha$ in REG acts as the natural counterpart to the expectile $\tau$, recovering nearly identical value estimates on synthetic tasks. This confirms that the success of IQL-style methods stems from their implicit solution to a robustness problem.

**Value Learning Objective.** Although the value learning objective above is self-contained, directly optimizing (8) can lead to unstable optimization when using in-sample estimated values for the Bellman backup (Lee et al., 2021). Previous studies identified that the forward gradient taken on the current state and the backward gradient taken on the next state may cancel out each other's effect (Mao et al., 2024b). To mitigate this issue, we adopt the stabilized actor–critic structure used in IQL (Kostrikov et al., 2021a). We introduce three parametrized functions: a critic $Q_\phi(s,a)$, a target network $Q_{\phi'}(s,a)$ and a state-value function $V_\psi(s)$. This decouples the learning process into two stable supervised objectives. The critic $Q_\phi(s,a)$ learns the noisy Bellman backup target by minimizing a standard mean square error (MSE) loss. This effectively averages the Bellman target over many gradient steps. The target network $Q_{\phi'}(s,a)$ is updated via Polyak (exponential) averaging. The value function $V_\psi(s)$ then learns from the lagging critic $Q_{\phi'}(s,a)$:

$$\mathcal{L}_\psi = \mathbb{E}_{(s,a)\sim\mathcal{D}}\left[\frac{1}{2}(\max(0, Q_{\phi'}(s,a) - V_\psi(s))^2\right] \tag{10}$$
$$+ \alpha\mathbb{E}_{s\sim\mathcal{D}}[V_\psi(s)]$$
$$\mathcal{L}_\phi = \mathbb{E}_{(s,a,s')\sim\mathcal{D}}\left[(r(s,a) + \gamma V_\psi(s') - Q_\phi(s,a))^2\right] \tag{11}$$

Note that, in the second term of (10), we extend the initial distribution $d_0$ to $\mathcal{D}$ in order to increase the sample diversity (Kostrikov et al., 2019; Nachum et al., 2019).

**Weighted Behavior Cloning (WBC).** Given the density ratio $\omega(s,a)$ recovered from the learned values (9), we can optimize a parametrized policy $\pi_\theta$ by maximizing a weighted log-likelihood $\max_\theta \mathbb{E}_{(s,a)\sim d_{\pi^*}}[\log\pi_\theta(a\mid s)]$ on state–action pairs sampled from the dataset with the learned importance-sampling weight $\omega$

$$\max_\theta \mathbb{E}_{(s,a)\sim\mathcal{D}}\left[\max(0, Q_\phi(s,a) - V_\psi(s))\log\pi_\theta(a\mid s)\right].$$

This update is inherently safe in the sense that it only queries the critic on in-sample actions. However, it has been noted that such forward KL-divergence objectives are mode-covering and can be overly conservative, potentially averaging over multiple good actions rather than committing to the best one (Park et al., 2024).

**Orthogonal Policy Gradients (OPG).** To enhance mode-seeking behavior, we require a more aggressive policy improvement mechanism. The canonical choice is an on-policy policy gradient objective. However, naively applying this is unsafe in the offline setting due to its reliance on querying the critic with OOD actions, risking exploitation of extrapolation errors.

We propose a method that regularizes the potentially unsafe on-policy policy gradient using the theoretically-grounded WBC gradient. Instead of a simple linear mixture, we use a geometric decomposition that we call orthogonal policy gradients (OPG). We define two gradient signals:

1. **The Conservative Gradient ($g_{\text{wbc}}$):** The gradient of the WBC objective. It is in-sample, stable, and provides a data-supported direction for improvement.

$$g_{\text{wbc}}(\theta) = \mathbb{E}_{(s,a)\sim\mathcal{D}}[\omega(s,a)\nabla_\theta\log\pi_\theta(a\mid s)]$$

2. **The Policy Gradient ($g_{\text{pg}}$):** The standard on-policy gradient. It is mode-seeking but potentially unstable.

$$g_{\text{pg}}(\theta) = \mathbb{E}_{s\sim\mathcal{D},a\sim\pi_\theta(\cdot|s)}[Q(s,a)\nabla_\theta\log\pi_\theta(a\mid s)]$$

The final gradient update is a careful combination of both:

$$g_{\text{final}}(\theta) = g_{\text{wbc}}(\theta) + \lambda\left(g_{\text{pg}}(\theta) - \text{proj}_{g_{\text{wbc}}}(g_{\text{pg}}(\theta))\right)$$
$$= g_{\text{wbc}}(\theta) + \lambda g_{\text{pg}}^\perp(\theta). \tag{12}$$

This update takes the full safe step $g_{\text{wbc}}$ and adds only the component of the policy gradient, $g_{\text{pg}}^\perp$, that is orthogonal to it. This ensures we only incorporate probing information to the extent that it provides a novel direction for improvement, not already captured by the conservative update. This formulation yields an emergent adaptive safety mechanism. When the policy gradient $g_{\text{pg}}$ aligns with the conservative gradient $g_{\text{wbc}}$, i.e., their inner product is positive, the update proceeds as a regularized policy improvement step. However, when the two gradients have conflicting directions, the update rule automatically pulls towards the safe, data-supported policy direction. This implicit "safety brake" allows the policy update to be aggressive when value improvement is safe and cautious when high-value regions are unsupported by data. As we show in the following experiments, OPG yields performance gains across almost all benchmark datasets, and we further analyze its sensitivity of the hyperparameter $\lambda$. The final method is presented in Algorithm 1.

**Algorithm 1** Regularized Evaluation Gap (REG)

1: **Initialize** critic $Q_\phi$, target critic $Q_{\phi'}$, value function $V_\psi$, and policy $\pi_\theta$
2: **repeat**: Sample mini-batch $\mathcal{B} = \{(\mathbf{s}, \mathbf{a}, r, \mathbf{s}')\} \sim \mathcal{D}$
3:      Update $V_\psi$ by minimizing (10) using $\mathcal{B}$
4:      Update $Q_\phi$ by minimizing (11) using $\mathcal{B}$
5:      Update target critic: $\phi' \leftarrow \beta\phi + (1 - \beta)\phi'$
6:      Update $\pi_\theta$ with gradient (12) using $\mathcal{B}$

# 5. Experiments

In this section, we empirically evaluate our proposed method, Regularized Evaluation Gap (REG) equipped with a standard Gaussian policy, in the standard offline RL setting. We conduct experiments on the widely-used D4RL benchmark suite and compare REG in Algorithm 1 with state-of-the-art prior methods. We then provide a targeted ablation study to analyze the contribution of our Orthogonal Policy Gradient (OPG) update rule. Full hyperparameters and additional experimental details are available in Appendix C.

## 5.1. Results on D4RL Benchmarks

We evaluate the performance of REG on the D4RL benchmark (Fu et al., 2020), including MuJoCo locomotion tasks and AntMaze navigation tasks. While MuJoCo tasks are prevalent in offline RL evaluation, AntMaze offline datasets are considered more challenging, possibly due to the tendency of subsequences of good behavior to be shorter, obstructing credit assignment. For baseline algorithms, we selected reputedly performant prior methods using either a Gaussian policy or more flexible diffusion policies. Gaussian policy baselines include IQL (Kostrikov et al., 2021b), IVR (Xu et al., 2023), AlignIQL (He et al., 2024), Extreme-QL (Garg et al., 2023), and ReBRAC (Tarasov et al., 2023). Diffusion policy baselines include Diffusion-QL (Wang et al., 2023) and IDQL (Hansen-Estruch et al., 2023).

The results presented in Table 1 demonstrate that REG is a highly effective offline RL algorithm. In comparison to methods that use a standard Gaussian policy, REG matches the state-of-the-art performance. We observe particularly strong performance on the noisy suboptimal datasets, such as `medium` and `medium-replay` (see Appendix C for data information). Success on these tasks, which require significant policy improvement over the dataset's behavior, suggests that our OPG update effectively balances in-sample safety with the aggressive mode-seeking needed for optimization. Furthermore, it is notable that REG achieves this strong performance without relying on complex generative models for the policy. It is equally competitive with Diffusion-QL and IDQL while using a simpler and more computationally efficient Gaussian policy architecture. This highlights the strength of our value regularization framework, which can

extract high-quality policies without the significant overhead of diffusion-based samplers. In summary, REG matches the state-of-the-art among methods with both simple policy architectures and more complex generative approaches, thanks to a synergy between its robust value learning and adaptive OPG policy update.

## 5.2. Ablation Study

In this section, we conduct a series of targeted experiments to investigate the key components and hyperparameter sensitivities of our REG framework. Specifically, we aim to: (1) examine the impact of $\alpha$, which governs the conservativeness of the value learning objective; (2) empirically validate the effectiveness of our Orthogonal Policy Gradient (OPG) update; and (3) analyze the role of the hyperparameter $\lambda$, which controls the strength of the exploratory gradient. Additional sensitivity analyses are provided in Appendix C.

**Sensitivity to $\alpha$.** We first examine the role of $\alpha$, which acts as a conservativeness controller, with higher $\alpha$ corresponding to more conservative value estimation. Figure 1 examines this trade-off. The optimal choice of $\alpha$ appears mildly task-dependent. Datasets in the categories `medium` and `medium-replay` exhibit a strong preference for higher $\alpha$ values, suggesting that a more conservative value learning objective is beneficial when the dataset is largely mixed of suboptimal demonstration. In contrast, for `expert` datasets, performance is more robust to the choice of $\alpha$ and shows a mild preference for lower $\alpha$. This highlights the role of $\alpha$ as a key knob for tuning the algorithm's conservativeness.

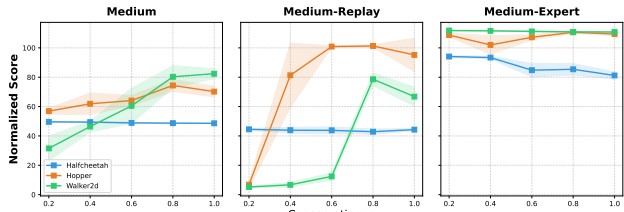

*Figure 1.* Sensitivity analysis for the REG hyperparameter $\alpha$, in terms of normalized D4RL scores averaged over 5 seeds with $\lambda = 0$.

**Effectiveness of Orthogonal Policy Gradient.** The OPG update is designed to bridge the gap between stable in-sample imitation and aggressive mode-seeking. To validate the practical utility of this design, we compare the performance of the REG method using only the theoretically-grounded WBC gradient with using the OPG gradient. Figure 2 presents the results aggregated over MuJoCo locomotion tasks across three dataset types. The results show that OPG consistently and significantly outperforms the WBC-only ablation. This performance gap is particularly pronounced in the `medium` and `medium-replay` datasets, where simple conservative imitation (WBC) is insufficient to recover the optimal

*Table 1.* **D4RL offline RL results.** Scores are averaged over the final 10 evaluations across 5 seeds with standard deviation reported. We highlight the top two scores among all methods ( **best result** , second best ). Parameters for each dataset are listed in Appendix C.

| Dataset | Gaussian Policy (Baselines) | | | | | Diffusion Policy | | Gaussian Policy |
|---|---|---|---|---|---|---|---|---|
| | IQL | IVR | AlignIQL | EQL | ReBRAC | Diffusion-QL | IDQL | REG (ours) |
| halfcheetah-medium | 47.4 ± 0.2 | 48.3 ± 0.2 | 44.2 ± 0.3 | 48.3 | **65.6 ±1.0** | 51.1 ± 0.5 | 51.0 | 53.6 ± 0.5 |
| hopper-medium | 66.3 ± 5.7 | 75.5 ± 3.4 | 57.8 ± 2.4 | 74.2 | **102.0 ±1.0** | 90.5 ± 4.6 | 65.4 | 91.9 ± 6.7 |
| walker2d-medium | 72.5 ± 8.7 | 84.2 ± 4.6 | 76.7 ± 3.4 | 84.2 | 82.5 ±3.6 | 87.0 ± 0.9 | 82.5 | **87.9 ± 1.7** |
| halfcheetah-medium-replay | 44.2 ± 1.2 | 44.8 ± 0.7 | 37.3 ± 0.2 | 45.2 | **51.2 ±3.2** | 47.8 ± 0.3 | 45.9 | 50.0 ± 0.2 |
| hopper-medium-replay | 95.2 ± 8.6 | 99.7 ± 3.3 | 77.9 ± 28.9 | 100.7 | 98.1 ±5.3 | 101.3 ± 0.6 | 92.1 | **101.4 ± 0.5** |
| walker2d-medium-replay | 76.1 ± 7.3 | 81.2 ± 3.8 | 66.3 ± 9.1 | 82.2 | 77.3 ±7.9 | **95.5 ± 1.5** | 85.1 | 87.3 ± 4.3 |
| halfcheetah-medium-expert | 86.7 ± 5.3 | 94.0 ± 0.4 | 81.9 ± 1.5 | 94.2 | **101.1 ±5.2** | 96.8 ± 0.3 | 95.9 | 94.1 ± 0.6 |
| hopper-medium-expert | 101.5 ± 7.3 | **111.8 ± 2.2** | 75.2 ± 5.9 | 111.2 | 107.0 ±6.4 | 111.1 ± 1.3 | 108.6 | 110.6 ± 0.9 |
| walker2d-medium-expert | 110.6 ± 1.0 | 110.0 ± 0.8 | 104.4 ± 9.5 | 112.7 | 111.6 ±0.3 | 110.1 ± 0.3 | 112.7 | **113.5 ± 0.8** |
| **Locomotion-Average** | 77.8 | 83.3 | 69.1 | 83.7 | 88.4 | 87.9 | 81.9 | 87.8 ± 0.9 |
| antmaze-umaze | 77.0 ± 5.5 | 92.2 ± 1.4 | 95.6 ± 2.2 | 93.8 | **97.8 ± 1.0** | 93.4 ± 3.4 | 94.0 | 94.1 ± 1.6 |
| antmaze-umaze-diverse | 54.3 ± 5.5 | 74.0 ± 2.3 | 72.0 ± 7.3 | 82.0 | **88.3 ±13.0** | 66.2 ± 8.6 | 80.2 | 76.4 ± 6.1 |
| antmaze-medium-play | 65.8 ± 11.7 | 80.2 ± 3.7 | **88.0 ± 2.7** | 76.0 | 84.0 ±4.2 | 76.6 ± 10.8 | 84.5 | 85.0 ± 5.2 |
| antmaze-medium-diverse | 73.8 ± 5.5 | 79.1 ± 4.2 | 83.2 ± 5.2 | 73.6 | 76.3 ±13.5 | 78.6 ± 10.3 | **84.8** | 84.4 ± 4.9 |
| antmaze-large-play | 42.0 ± 4.5 | 53.2 ± 4.8 | 55.2 ± 9.5 | 46.5 | 60.4 ±26.1 | 46.4 ± 8.3 | 63.5 | **63.6 ± 8.6** |
| antmaze-large-diverse | 30.3 ± 3.6 | 52.3 ± 5.2 | 58.0 ± 3.6 | 49.0 | 54.4 ±25.1 | 56.6 ± 7.6 | **67.9** | 61.2 ± 5.2 |
| **AntMaze-Average** | 57.2 | 71.8 | 75.3 | 70.2 | 76.8 | 69.6 | 79.2 | 77.5 ± 2.3 |

policy. The substantial improvement from OPG empirically confirms that the orthogonal mode-seeking component can successfully guide the policy toward higher-reward modes that a purely conservative update may miss. Furthermore, the sustained performance throughout the various experiments provides empirical support for the stability mechanism: despite potentially unsafe mode-seeking, the WBC gradient prevents the policy from collapsing due to OOD extrapolation. In `medium-expert` datasets, the improvement is modest, as the near-optimal behavior data already allows WBC to perform well on its own.

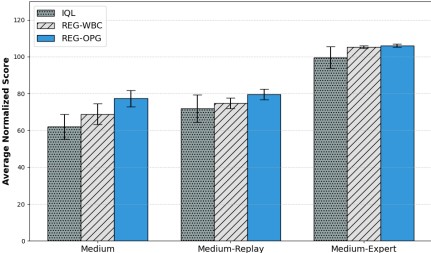

*Figure 2.* Comparison of the WBC and OPG modes of REG, in terms of average normalized scores over different dataset types, with 95% confidence intervals.

**Sensitivity to Mode-Seeking Gradient Strength** ($\lambda$). The hyperparameter $\lambda$ scales the orthogonal policy gradient $g_{\text{pg}}^{\perp}$, directly controlling the trade-off between the conservative WBC anchor and aggressive mode-seeking. Figure 3 illustrates performance across a range of $\lambda$ values. In most `medium` and `medium-replay` datasets, we observe a unimodal tendency, with performance peaking at an intermediate value of $\lambda \in [0.2, 0.5]$ and decreasing for lower and higher values. Excessively large $\lambda$ values can lead to instability and performance degradation, particularly on `expert`-level datasets where the aggressive gradient signal

has little benefit over the high-quality data. This finding suggests that, while orthogonal exploration is beneficial, it must be carefully balanced against safe, data-supported directions provided by the conservative gradient. The hyperparameter $\lambda$ serves as the primary knob for controlling this balance.

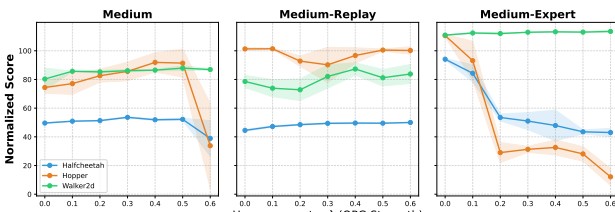

*Figure 3.* Sensitivity analysis for the OPG hyperparameter $\lambda$. Scores are normalized D4RL scores, averaged over 5 seeds.

## 6. Conclusion

In this paper, we introduced the Regularized Evaluation Gap (REG) framework, which provides a principled robust optimization perspective on offline RL and establishes a link to existing implicit value learning algorithms. Based on this formulation, we developed a practical policy extraction method that leverages robust value estimation via the Orthogonal Policy Gradient (OPG) technique. Our experiments on the D4RL benchmark suite demonstrate that REG matches the state-of-the-art performance of both Gaussian-policy methods and more computationally intensive diffusion-based approaches. Future directions include extending REG to incorporate more expressive policy representations, such as flow-based (McAllister et al., 2025) or diffusion models (Fang et al., 2025), and scaling the framework to off-policy online RL settings (Xu et al., 2025).

## Acknowledgments

This work was funded in part by the National Science Foundation (Award #2321786). We thank the anonymous reviewers and area chair for their constructive feedback that improved the clarity and rigor of this paper.

## Impact Statement

This paper introduces a framework for robust offline reinforcement learning. By enabling agents to learn safely from fixed datasets without requiring online interaction, our method potentially lowers the barrier to deploying RL in high-stakes domains such as healthcare and robotics, where trial-and-error can be unsafe. While the general risks of autonomous decision-making systems apply, this work does not introduce specific ethical concerns beyond those standard to the field.

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

# A. Proofs

## A.1. Proof of Theorem 4.2

To prove the generalized dual formulation, we first update the equivalence between absolute and one-sided constraints using the skew indicator $\kappa$, followed by the duality derivation.

Define the linear operator $\mathcal{P}^\pi : \mathbb{R}^{|\mathcal{S}||\mathcal{A}|} \to \mathbb{R}^{|\mathcal{S}||\mathcal{A}|}$ by

$$(\mathcal{P}^\pi q)(s, a) := \mathbb{E}_{s' \sim P(\cdot|s,a),\, a' \sim \pi(\cdot|s')}[q(s', a')].$$

Then the Bellman residual can be written as

$$\Delta_\pi q \;=\; r + \gamma \mathcal{P}^\pi q - q.$$

**Lemma A.1** (Residual realizability). *For any fixed $\pi$, the mapping $q \mapsto \Delta_\pi q$ is an affine bijection on $\mathbb{R}^{|\mathcal{S}||\mathcal{A}|}$. In particular, for any residual function $\Delta : \mathcal{S} \times \mathcal{A} \to \mathbb{R}$, there exists a unique $q$ such that $\Delta = \Delta_\pi q$, given by*

$$q = (I - \gamma\mathcal{P}^\pi)^{-1}(r - \Delta) \quad where \quad (I - \gamma\mathcal{P}^\pi)^{-1} = \sum_{t=0}^\infty (\gamma\mathcal{P}^\pi)^t.$$

*Proof.* From $\Delta_\pi q = r + \gamma\mathcal{P}^\pi q - q$, we have $(I - \gamma\mathcal{P}^\pi)q = r - \Delta_\pi q$. Since $\mathcal{P}^\pi$ is a stochastic linear operator in the finite case, $\|\mathcal{P}^\pi\|_\infty \le 1$, and thus the Neumann series $\sum_{t \ge 0}(\gamma\mathcal{P}^\pi)^t$ converges and equals $(I - \gamma\mathcal{P}^\pi)^{-1}$. Therefore for any $\Delta$ the equation $(I - \gamma\mathcal{P}^\pi)q = r - \Delta$ has a unique solution, proving bijectivity. □

**Lemma A.2.** *Let $f : \mathbb{R} \to \mathbb{R}_+$ be convex with $f(0) = 0$. Define $\kappa \in \{1, -1\}$ such that $f(\kappa x) \le f(-\kappa x)$ for all $x > 0$. Then:*

$$\sup_{q \in Q(\pi)} \left|\mathbb{E}_{d_\pi}[\Delta_\pi q(s, a)]\right| = \sup_{q \in Q(\pi)} \mathbb{E}_{d_\pi}[\kappa \cdot \Delta_\pi q(s, a)].$$

*Proof.* Let $Val(\pi) := \max_{q \in Q(\pi)}\left|\mathbb{E}_{d_\pi}[\Delta_\pi q(s, a)]\right|$. By definition of the absolute value, we decompose the maximization into two one-sided optimization problems based on the direction of $\kappa$:

$$Val(\pi) = \max\left\{ \underbrace{\max_{q \in Q(\pi)} \mathbb{E}_{d_\pi}[\kappa \cdot \Delta_\pi q(s, a)]}_{P_{\text{aligned}}}, \quad \underbrace{\max_{q \in Q(\pi)} \mathbb{E}_{d_\pi}[-\kappa \cdot \Delta_\pi q(s, a)]}_{P_{\text{opposed}}} \right\}.$$

We aim to prove that $P_{\text{aligned}} \ge P_{\text{opposed}}$, which implies $Val(\pi) = P_{\text{aligned}}$.

Let $q^*$ be a maximizer of the opposing problem $P_{\text{opposed}}$. We define its residual as $\Delta^*(s, a) := \Delta_\pi q^*(s, a)$. We first argue that the quantity we are maximizing, $-\kappa\Delta^*(s, a)$, must be non-negative for almost all $(s, a)$. Suppose for the sake of contradiction that $-\kappa\Delta^*$ is negative on a set of non-zero measure. Construct a rectified residual $\Delta'$ such that:

$$-\kappa\Delta'(s, a) = \max(0, -\kappa\Delta^*(s, a)).$$

- **Objective Improvement:** Since we replaced negative contributions with zero, the sum increases: $\mathbb{E}_{d_\pi}[-\kappa\Delta'] > \mathbb{E}_{d_\pi}[-\kappa\Delta^*]$.

- **Constraint Satisfaction:** Since $f$ is convex with a minimum at $f(0) = 0$, moving a value closer to 0 reduces cost. Thus, $f(\Delta') \le f(\Delta^*)$, ensuring $\Delta'$ remains feasible.

Therefore, the optimal residual for $P_{\text{opposed}}$ must satisfy $-\kappa\Delta^*(s, a) \ge 0$ almost everywhere. Let $y(s, a) = -\kappa\Delta^*(s, a) \ge 0$. Then $\Delta^*(s, a) = -\kappa y(s, a)$.

We now construct a candidate residual for the aligned problem $P_{\text{aligned}}$, defined as $\Delta_{\text{new}}(s, a) := -\Delta^*(s, a) = \kappa y(s, a)$. First, we verify feasibility using the skew property of $\kappa$. By definition, $\kappa$ points to the cheaper direction, meaning $f(\kappa y) \le f(-\kappa y)$ for any $y \ge 0$.

$$\mathbb{E}_{d_\mu}[f(\Delta_{\text{new}})] = \mathbb{E}_{d_\mu}[f(\kappa y)] \le \mathbb{E}_{d_\mu}[f(-\kappa y)] = \mathbb{E}_{d_\mu}[f(\Delta^*)] \le \epsilon_2.$$

Thus, the critic $q_{\text{new}}$ corresponding to $\Delta_{\text{new}}$ is feasible.

Next, we evaluate the objective of this candidate for $P_{\text{aligned}}$:

$$\mathbb{E}_{d_\pi}[\kappa \cdot \Delta_{\text{new}}] = \mathbb{E}_{d_\pi}[\kappa(\kappa y)] = \mathbb{E}_{d_\pi}[y] = \mathbb{E}_{d_\pi}[-\kappa \Delta^*] = P_{\text{opposed}}.$$

Since $P_{\text{aligned}}$ is the maximum over the set, and we found a feasible candidate achieving the value $P_{\text{opposed}}$, it must be that $P_{\text{aligned}} \geq P_{\text{opposed}}$.

Substituting this result back into the decomposition:

$$Val(\pi) = \max\{P_{\text{aligned}}, P_{\text{opposed}}\} = P_{\text{aligned}} = \max_{q \in Q(\pi)} \mathbb{E}_{d_\pi}[\kappa \cdot \Delta_\pi q(s, a)].$$

$\square$

**Lemma A.3.** *The constrained objective is equivalent to the regularized optimization problem:*

$$\max_{\pi \in \Pi_{\text{in}}} \min_{\alpha_1 \geq 0} \max_{\alpha_2 \geq 0} \quad \mathbb{E}_{d_\pi}[r(s, a)] + \alpha_1 \epsilon_1 - \alpha_1 \alpha_2 \epsilon_2 - \alpha_1 \alpha_2 \mathbb{E}_{d_\mu}\left[f^*\left(\frac{\kappa d_\pi(s, a)}{\alpha_2 d_\mu(s, a)}\right)\right]$$

*subject to the flow constraints on $d_\pi$.*

*Proof.* The proof relies on Fenchel duality applied to the robust constraint. From Lemma A.2, the inner optimization problem characterizing the worst-case evaluation error is:

$$\max_{q \in Q(\pi)} \mathbb{E}_{d_\pi}[\kappa \cdot \Delta_\pi q(s, a)] \leq \epsilon_1$$

Using Lemma A.1, we may equivalently optimize over residuals $\Delta_\pi q$. Introduce a Lagrange multiplier $\alpha_1 \geq 0$ for this outer constraint, yielding the Lagrangian form

$$\max_{\Delta_\pi q} \min_{\alpha_2 \geq 0} \left(\mathbb{E}_{d_\pi}[\kappa \Delta_\pi q] - \alpha_2(\mathbb{E}_{d_\mu}[f(\Delta_\pi q)] - \epsilon_2)\right)$$

Now handle the inner supremum by Lagrangian duality with multiplier $\alpha_2 \geq 0$ for the ambiguity constraint. By the Slater condition, convexity, strong duality applies and using coverage $d_\pi \ll d_\mu$ we can write

$$\min_{\alpha_2 \geq 0} \left(\alpha_2 \epsilon_2 + \max_{\Delta_\pi q} \mathbb{E}_{d_\mu}\left[\frac{d_\pi}{d_\mu} \kappa \Delta_\pi q - \alpha_2 f(\Delta_\pi q)\right]\right)$$

By the definition of the Fenchel conjugate,

$$\sup_{x \in \mathbb{R}}\{yx - \alpha_2 f(x)\} = \alpha_2 f^*(y/\alpha_2).$$

Applying this pointwise with $y = \frac{\kappa d_\pi(s, a)}{d_\mu(s, a)}$ yields

$$\sup_{\Delta_\pi q}\left(\Delta_\pi q\left(\frac{\kappa d_\pi(s, a)}{d_\mu(s, a)}\right) - \alpha_2 f(\Delta_\pi q)\right) = \alpha_2 \sup_{\Delta_\pi q}\left(\Delta_\pi q\left(\frac{\kappa d_\pi(s, a)}{\alpha_2 d_\mu(s, a)}\right) - f(\Delta_\pi q)\right)$$

$$= \alpha_2 f^*\left(\frac{\kappa d_\pi(s, a)}{\alpha_2 d_\mu(s, a)}\right)$$

Thus, the value of the inner optimization is:

$$\min_{\alpha_2 \geq 0} \alpha_2 \mathbb{E}_{d_\mu}\left[f^*\left(\frac{\kappa d_\pi(s, a)}{\alpha_2 d_\mu(s, a)}\right)\right] + \alpha_2 \epsilon_2$$

Finally, we convert the robust constraint into a regularizer using a Lagrange multiplier $\alpha_1 \geq 0$. Since the term enters negatively in the maximization objective for $\pi$, the minimization over $\alpha_2$ becomes a maximization. This yields the stated result. $\square$

**Proof of Theorem 4.2**

*Proof.* We work in the discounted occupancy-measure formulation: optimize over $d_\pi$ which by definition follows the Bellman flow and $d_\pi \geq 0$. For fixed $(\alpha_1, \alpha_2)$, the objective in Lemma A.3 is concave in $d_\pi$ (it is linear in $d_\pi$ minus a convex function of $d_\pi$), and the constraints are linear; hence this is a finite-dimensional convex optimization problem. By the Slater condition, strong duality applies. Thus we may introduce Lagrange multipliers $V(s)$ for the flow constraints and write the saddle formulation

$$\min_{\alpha_1 \geq 0} \max_{\alpha_2 \geq 0} \min_V \max_{d_\pi \geq 0} \left\{ \mathcal{L}(d_\pi, \alpha_1, \alpha_2) + \sum_s V(s) \left[ \gamma \sum_{s', a'} d_\pi(s', a') P(s \mid s', a') + (1 - \gamma) d_0(s) - \sum_a d_\pi(s, a) \right] \right\},$$

where $\mathcal{L}$ is as in Lemma A.3:

$$\mathcal{L}(d_\pi, \alpha_1, \alpha_2) = \mathbb{E}_{d_\pi}[r(s, a)] + \alpha_1 \epsilon_1 - \alpha_1 \alpha_2 \epsilon_2 - \alpha_1 \alpha_2 \mathbb{E}_{d_\mu}\left[ f^*\left( \frac{\kappa d_\pi(s, a)}{\alpha_2 d_\mu(s, a)} \right) \right].$$

Now peel the onion from inside out. Collecting the terms that multiply $d_\pi(s, a)$ defines

$$A(s, a) := r(s, a) + \gamma \sum_{s'} V(s') P(s' \mid s, a) - V(s).$$

The inner maximization over $d_\pi \geq 0$ becomes

$$\max_{d_\pi \geq 0} \mathbb{E}_{d_\pi}[A(s, a)] - \alpha_1 \alpha_2 \mathbb{E}_{d_\mu}\left[ f^*\left( \frac{\kappa d_\pi}{\alpha_2 d_\mu} \right) \right] + (1 - \gamma) \mathbb{E}_{d_0}[V(s)].$$

Using coverage $d_\pi \ll d_\mu$, change variables to the occupancy ratio

$$\omega(s, a) := \frac{d_\pi(s, a)}{d_\mu(s, a)} \quad (d_\mu\text{-a.e.}), \qquad \omega(s, a) \geq 0.$$

Then $\mathbb{E}_{d_\pi}[A] = \mathbb{E}_{d_\mu}[A \, \omega]$ and the inner problem is

$$\min_V \max_{\omega \geq 0} \mathbb{E}_{d_\mu}\left[ A(s, a)\omega(s, a) - \alpha_1 \alpha_2 f^*\left( \frac{\kappa \omega(s, a)}{\alpha_2} \right) \right] + (1 - \gamma) \mathbb{E}_{d_0}[V(s)].$$

This maximization decouples pointwise over $(s, a)$. Write $y := \frac{\kappa \omega}{\alpha_2}$; since $\omega \geq 0$, we have $y \in \kappa \mathbb{R}_+$. Then for each $(s, a)$,

$$A\omega - \alpha_1 \alpha_2 f^*\left( \frac{\kappa \omega}{\alpha_2} \right) = \alpha_1 \alpha_2 \left[ \left( \frac{\kappa A}{\alpha_1} \right) y - f^*(y) \right].$$

We use the following one-sided Fenchel identity:

**Claim.** For any closed convex $f : \mathbb{R} \to \mathbb{R}_+$ with $f(0) = 0$ and any $\kappa \in \{\pm 1\}$,

$$\sup_{y \in \kappa \mathbb{R}_+} \{xy - f^*(y)\} = \mathbb{I}\{\kappa x \geq 0\} f(x).$$

*Justification.* If $\kappa x < 0$, then $xy \leq 0$ for all $y \in \kappa \mathbb{R}_+$ and $f^*(y) \geq 0$ with equality at $y = 0$, so the supremum is 0. If $\kappa x \geq 0$, the unconstrained supremum over $y \in \mathbb{R}$ equals $f(x)$ by Fenchel conjugacy, and it is attained by some $y \in \partial f(x)$; since $f$ is convex and minimized at 0, subgradients have the same sign as $x$, hence such a maximizer lies in $\kappa \mathbb{R}_+$ and the restricted supremum also equals $f(x)$. $\square$

Applying the claim with $x = \frac{\kappa A}{\alpha_1}$ gives $\kappa x = A/\alpha_1$, hence

$$\sup_{\omega \geq 0} \left\{ A\omega - \alpha_1 \alpha_2 f^*\left( \frac{\kappa \omega}{\alpha_2} \right) \right\} = \alpha_1 \alpha_2 \, \mathbb{I}\{A \geq 0\} \, f\left( \frac{\kappa A}{\alpha_1} \right).$$

Therefore the inner optimization reduces to

$$\min_{V(s)} \mathbb{E}_{d_\mu}\left[\alpha_1\alpha_2 \,\mathbb{I}\{A(s,a) \ge 0\}\, f\!\left(\frac{\kappa A(s,a)}{\alpha_1}\right)\right] + (1-\gamma)\mathbb{E}_{d_0}[V(s)].$$

Substituting back yields the final saddle problem

$$\min_{\alpha_1 \ge 0} \max_{\alpha_2 \ge 0} \min_{V(s)} \quad \mathbb{E}_{d_\mu}\left[\alpha_1\alpha_2 \,\mathbb{I}\{A(s,a) \ge 0\}\, f\!\left(\frac{\kappa A(s,a)}{\alpha_1}\right)\right] + (1-\gamma)\mathbb{E}_{d_0}[V(s)] + \alpha_1\epsilon_1 - \alpha_1\alpha_2\epsilon_2.$$

Finally, the optimal ratio $\omega^*(s,a) = \frac{d_{\pi^*}(s,a)}{d_\mu(s,a)}$ is obtained from the pointwise maximization. Whenever $A(s,a) < 0$, the above derivation implies $\omega^*(s,a) = 0$. Whenever $A(s,a) \ge 0$, an optimizer $y^*(s,a)$ satisfies the (restricted) Fenchel optimality condition

$$y^*(s,a) \in \arg\max_{y \in \kappa\mathbb{R}_+}\left\{\left(\frac{\kappa A(s,a)}{\alpha_1}\right)y - f^*(y)\right\}, \quad \text{so} \quad y^*(s,a) \in \partial f\!\left(\frac{\kappa A(s,a)}{\alpha_1}\right) \cap \kappa\mathbb{R}_+,$$

and thus

$$\omega^*(s,a) = \alpha_2\,\kappa\,y^*(s,a).$$

If in addition $f$ is differentiable, then $y^* = f'(\kappa A/\alpha_1)$ and we obtain the closed form

$$\omega^*(s,a) = \max\left\{0,\ \alpha_2\,\kappa\,f'\!\left(\frac{\kappa A(s,a)}{\alpha_1}\right)\right\}.$$

$\square$

## A.2. Proof of Lemma 4.1

*Proof of Evaluation Lemma.* Starting from the definition of the expected Bellman error for a critic $q(s,a)$ under policy $\pi$, we expand the expectation using the visitation distribution $d_\pi$:

$$\mathbb{E}_{d_\pi}[\Delta_\pi q(s,a)] = \mathbb{E}_{d_\pi}\left[r(s,a) + \gamma \sum_{s',a'} P(s',a' \mid s,a)q(s',a') - q(s,a)\right]$$

$$= \underbrace{\sum_{s,a} d_\pi(s,a)r(s,a)}_{(1-\gamma)J_\pi} + \gamma \sum_{s,a} d_\pi(s,a) \sum_{s',a'} P(s',a' \mid s,a)q(s',a') - \sum_{s,a} d_\pi(s,a)q(s,a).$$

We rearrange the second term by swapping the order of summation:

$$\mathbb{E}_{d_\pi}[\Delta_\pi q(s,a)] = (1-\gamma)J_\pi + \sum_{s',a'} q(s',a')\left(\gamma \sum_{s,a} d_\pi(s,a)P(s',a' \mid s,a)\right) - \sum_{s,a} d_\pi(s,a)q(s,a).$$

We now apply the definition of the visitation measure $d_\pi$, which states that for any $(s',a')$:

$$d_\pi(s',a') = (1-\gamma)d_0(s')\pi(a' \mid s') + \gamma \sum_{s,a} d_\pi(s,a)P(s',a' \mid s,a).$$

Substituting the term in the parenthesis with $d_\pi(s',a') - (1-\gamma)d_0(s')\pi(a' \mid s')$, we obtain:

$$\mathbb{E}_{d_\pi}[\Delta_\pi q(s,a)] = (1-\gamma)J_\pi + \sum_{s',a'} q(s',a')[d_\pi(s',a') - (1-\gamma)d_0(s')\pi(a' \mid s')] - \sum_{s,a} d_\pi(s,a)q(s,a)$$

$$= (1-\gamma)J_\pi + \sum_{s',a'} d_\pi(s',a')q(s',a') - (1-\gamma)\sum_{s',a'} d_0(s')\pi(a' \mid s')q(s',a') - \sum_{s,a} d_\pi(s,a)q(s,a).$$

Observing that the terms $\sum_{s',a'} d_\pi(s',a')q(s',a')$ and $\sum_{s,a} d_\pi(s,a)q(s,a)$ cancel out, we are left with:

$$\mathbb{E}_{d_\pi}[\Delta_\pi q(s,a)] = (1-\gamma)J_\pi - (1-\gamma)\mathbb{E}_{s\sim d_0,a\sim\pi}[q(s,a)].$$

Rearranging the terms yields the final evaluation gap identity:

$$J_\pi - \mathbb{E}_{s\sim d_0,a\sim\pi}[q(s,a)] = \frac{1}{1-\gamma}\mathbb{E}_{d_\pi}[\Delta_\pi q(s,a)]. \tag{13}$$

$\square$

### A.3. Proof of Theorem 4.6

**Lemma A.4.** *Let $(V^*, \omega^*, \alpha_1^*, \alpha_2^*)$ be an optimal saddle-point solution associated with Theorem 4.2. Let $A_{V^*}(s, a)$ be the advantage under $V^*$. Let $\Delta^*(s, a)$ be a worst-case Bellman residual attaining the inner supremum in the primal ambiguity set. Assume $f$ is closed, convex, with minimum at $f(0) = 0$. Let $\kappa \in \{1, -1\}$ be the skew indicator.*

*Then on the active region $\{(s, a) : A_{V^*}(s, a) \geq 0\}$, we have $\kappa \Delta^*(s, a) \geq 0$ and*

$$\Delta^*(s, a) = \Theta(\kappa A_{V^*}(s, a))$$

*Proof.* The proof follows from the Fenchel optimality conditions at the saddle point. In the proof of Theorem 4.2, $\omega(s, a) \geq 0$ is chosen pointwise to maximize

$$\phi(\omega) = A_{V^*}(s, a)\, \omega - \alpha_1^* \alpha_2^* f^*\left(\frac{\kappa \omega}{\alpha_2^*}\right), \qquad \omega \geq 0.$$

Let $y^*(s, a) := \frac{\kappa \omega^*(s,a)}{\alpha_2^*} \in \kappa \mathbb{R}_+$. The restricted Fenchel argument used in Theorem 4.2 implies that on the active region $A_{V^*}(s, a) \geq 0$, an optimizer satisfies

$$y^*(s, a) \in \partial f\left(\frac{\kappa A_{V^*}(s, a)}{\alpha_1^*}\right) \cap \kappa \mathbb{R}_+. \tag{14}$$

The appearance of $f^*$ comes from the pointwise conjugacy

$$f^*(y) = \sup_{\Delta \in \mathbb{R}}\{y\Delta - f(\Delta)\}.$$

Since $\Delta^*(s, a)$ attains this supremum at $y = y^*(s, a)$, Fenchel equality yields

$$y^*(s, a) \in \partial f\left(\Delta^*(s, a)\right) \iff \Delta^*(s, a) \in \partial f^*\left(y^*(s, a)\right). \tag{15}$$

Combining (14) and (15), we obtain that on $\{A_{V^*} \geq 0\}$,

$$y^*(s, a) \in \partial f\left(\frac{\kappa A_{V^*}(s, a)}{\alpha_1^*}\right) \quad \text{and} \quad y^*(s, a) \in \partial f\left(\Delta^*(s, a)\right).$$

Since $f$ is strictly convex, $\partial f$ is injective, hence the two arguments must coincide. In particular, $\kappa \Delta^*(s, a) \geq 0$ on the active region, and the displayed identity implies.

$$\Delta^*(s, a) = \Theta(\kappa A_{V^*}(s, a)) \qquad \text{whenever } A_{V^*}(s, a) \geq 0.$$

$\square$

**Theorem A.5** (Optimality Guarantee). *Assume the following conditions hold:*

1. *The convex loss function $f$ is L-smooth.*

2. *The $\epsilon_1$ such that the optimal policy $\pi^*$ is feasible $\alpha_1 \in [a^{-1}, a]$ for some constant $a \geq 1$.*

3. *The value function class $\mathcal{F}_V$ is bounded by $B$ (i.e., $\sup_s |V(s)| \leq B$) and has Rademacher complexity $\mathcal{R}_n(\mathcal{F}_V)$.*

*Then, with probability at least $1 - \delta$,*

$$J_{\pi^*} - J_{\hat{\pi}} \leq \frac{1}{1 - \gamma} O\left(\epsilon_1 + \alpha_2 \epsilon_2 + \alpha_2 L (R + 2B)^2 \sqrt{\frac{\ln(1/\delta)}{n}} + \alpha_2 L (R + 2B) \mathcal{R}_n(\mathcal{F}_V)\right)$$

*Proof.* Throughout, define the advantage under $V$ by

$$A_V(s, a) := r(s, a) + \gamma \mathbb{E}[V(s') \mid s, a] - V(s),$$

We decompose the suboptimality gap using the optimal primal value function $V^*$ and the learned value function $\hat{V}$:

$$J_{\pi^*} - J_{\hat{\pi}} = \underbrace{J_{\pi^*} - \mathbb{E}_{d_0}[V^*(s_0)]}_{\text{(A)}} + \underbrace{\mathbb{E}_{d_0}[V^*(s_0)] - \mathbb{E}_{d_0}[\hat{V}(s_0)]}_{\text{(B)}} + \underbrace{\mathbb{E}_{d_0}[\hat{V}(s_0)] - J_{\hat{\pi}}}_{\text{(C)}}.$$

**Term (A).** Starting from the definition of the advantage function, we have:

$$\mathbb{E}_{d_\pi}[A_V(s,a)] = \mathbb{E}_{d_\pi}\left[r(s,a) + \gamma \sum_{s'} P(s' \mid s,a)V(s') - V(s)\right]$$

$$= (1-\gamma)J_\pi + \gamma \sum_{s,a} d_\pi(s,a) \sum_{s'} P(s' \mid s,a)V(s') - \sum_{s,a} d_\pi(s,a)V(s)$$

$$= (1-\gamma)J_\pi + \gamma \sum_{s'} V(s') \sum_{s,a} d_\pi(s,a)P(s' \mid s,a) - \sum_s V(s) \sum_a d_\pi(s,a)$$

Using the Bellman flow constraint for the visitation measure $d_\pi$, $\sum_a d_\pi(s,a) = (1-\gamma)d_0(s) + \gamma \sum_{s',a'} d_\pi(s',a')P(s \mid s',a')$, we can write the second term:

$$\mathbb{E}_{d_\pi}[A_V(s,a)] = (1-\gamma)J_\pi + \sum_{s'} V(s')\left[\sum_a d_\pi(s',a) - (1-\gamma)d_0(s')\right] - \sum_s V(s) \sum_a d_\pi(s,a)$$

$$= (1-\gamma)J_\pi + \sum_{s'} V(s') \sum_a d_\pi(s',a) - (1-\gamma) \sum_{s'} V(s')d_0(s') - \sum_s V(s) \sum_a d_\pi(s,a)$$

$$= (1-\gamma)J_\pi - (1-\gamma)\mathbb{E}_{s\sim d_0}[V(s)]$$

Rearrange give us the Evaluation Gap under misspecified $V$:

$$J_\pi - \mathbb{E}_{d_0}[V(s)] = \frac{1}{1-\gamma}\mathbb{E}_{d_\pi}[A_V(s,a)] \tag{16}$$

Let $\Delta^*(s,a)$ denote the Bellman residual of the implicit worst-case critic in the primal robust constraint for $\pi^*$. Since $\pi^*$ is the solution to the primal problem, it must be feasible, the robust constraint implies:

$$\mathbb{E}_{d_{\pi^*}}[\Delta^*(s,a)] \le \epsilon_1$$

From the dual optimality conditions in Lemma A.4 and the condition of $\epsilon_1$, substituting this into the expectation we conclude.

$$(A) = J_{\pi^*} - \mathbb{E}_{d_0}[V^*] = \frac{1}{1-\gamma}\mathbb{E}_{d_{\pi^*}}[A_{V^*}(s,a)] \le O\left(\frac{\epsilon_1}{1-\gamma}\right)$$

**For term (B)** Let $L(V) = \mathbb{E}_{d_\mu}[g_V(s,a)] + (1-\gamma)\mathbb{E}_{s\sim d_0}[V(s)]$, $g_V(s,a) := \frac{\alpha_1\alpha_2}{2}\mathbb{I}(A_V(s,a) \ge 0)f\left(\frac{\kappa A_V(s,a)}{\alpha_1}\right)$. And its empirical counterpart $\hat{L}(V)$ defined by replacing $\mathbb{E}_{d_\mu}$ with the sample average $\hat{\mathbb{E}}_{d_\mu}$. Recall $\hat{V} \in \arg\min_{V\in\mathcal{F}_V} \hat{L}(V)$ and $V^* \in \arg\min_{V\in\mathcal{F}_V} L(V)$. By optimality of $V^*$ for $L$ and $\hat{V}$ for $\hat{L}$,

$$L(V^*) \le L(\hat{V}) \quad \text{and} \quad \hat{L}(\hat{V}) \le \hat{L}(V^*).$$

Then, using $L(V^*) \le L(\hat{V})$ and $\hat{L}(\hat{V}) \le \hat{L}(V^*)$, we have

$$\mathbb{E}_{d_0}[V^*(s_0) - \hat{V}(s_0)] \le \frac{1}{1-\gamma}\left(\mathbb{E}_{d_\mu}[g_{\hat{V}}] - \mathbb{E}_{d_\mu}[g_{V^*}]\right) \tag{17}$$

$$= \frac{1}{1-\gamma}\left[\left(\mathbb{E}_{d_\mu}[g_{\hat{V}}] - \hat{\mathbb{E}}_{d_\mu}[g_{\hat{V}}]\right) + \left(\hat{\mathbb{E}}_{d_\mu}[g_{\hat{V}}] - \hat{\mathbb{E}}_{d_\mu}[g_{V^*}]\right) + \left(\hat{\mathbb{E}}_{d_\mu}[g_{V^*}] - \mathbb{E}_{d_\mu}[g_{V^*}]\right)\right] \tag{18}$$

$$\le \frac{2}{1-\gamma}\sup_{V\in\mathcal{F}_V}|\mathbb{E}_{d_\mu}[g_V] - \hat{\mathbb{E}}_{d_\mu}[g_V]| + \hat{\mathbb{E}}_{d_\mu}[g_{\hat{V}}] - \hat{\mathbb{E}}_{d_\mu}[g_{V^*}] \tag{19}$$

Under the assumptions that the value function class $\mathcal{F}_V$ is some finite Rademacher complexity $\mathcal{R}_n(\mathcal{F}_V)$, for example reproducing kernel Hilbert space (RKHS) that are bounded by $B$. We can derive a uniformly convergence bound:

$$|\mathbb{E}_{d_\mu}[g_V] - \hat{\mathbb{E}}_{d_\mu}[g_V]| \le \alpha_2 L(R + 2B)\mathcal{R}_n(\mathcal{F}_V) + \alpha_2 L(R + 2B)^2\sqrt{\frac{\ln(2/\delta)}{2n}} \tag{20}$$

The proof of the uniform convergence comes with two-fold:

1. Bounding the Rademacher complexity of the empirical advantage function class

2. Applying Talagrand's Contraction Lemma

First, let $\mathcal{A}_{class} = \{A_V : (s, a, s') \mapsto A_V(s, a, s') \mid V \in \mathcal{F}_V\}$, where $A_V(s, a, s') = r(s, a) + \gamma V(s') - V(s)$. Then the Rademacher complexity of this class, for a fixed sample $D_n = \{(s_i, a_i, s'_i)\}_{i=1}^n$, is defined as:

$$\hat{\mathcal{R}}_{D_n}(\mathcal{A}_{class}) = \mathbb{E}_\sigma \left[ \sup_{V \in \mathcal{F}_V} \frac{1}{n} \sum_{i=1}^n \sigma_i A_V(s_i, a_i, s'_i) \right]$$

$$= \mathbb{E}_\sigma \left[ \sup_{V \in \mathcal{F}_V} \frac{1}{n} \sum_{i=1}^n \sigma_i (r(s_i, a_i) + \gamma V(s'_i) - V(s_i)) \right]$$

Since the term $\sum_{i=1}^n \sigma_i r(s_i, a_i)$ doesn't depend on $V$, we can separate it within the supremum:

$$\hat{\mathcal{R}}_{D_n}(\mathcal{A}_{class}) = \mathbb{E}_\sigma \left[ \frac{1}{n} \sum_{i=1}^n \sigma_i r(s_i, a_i) + \sup_{V \in \mathcal{F}_V} \frac{1}{n} \sum_{i=1}^n \sigma_i (\gamma V(s'_i) - V(s_i)) \right]$$

$$= \frac{1}{n} \sum_{i=1}^n r(s_i, a_i) \mathbb{E}_\sigma [\sigma_i] + \mathbb{E}_\sigma \left[ \sup_{V \in \mathcal{F}_V} \frac{1}{n} \left( \sum_{i=1}^n \sigma_i \gamma V(s'_i) - \sum_{i=1}^n \sigma_i V(s_i) \right) \right]$$

As $\mathbb{E}_\sigma [\sigma_i] = 0$, the first term vanishes. Using the sub-additivity of supremum, we can get

$$\hat{\mathcal{R}}_{D_n}(\mathcal{A}_{class}) \leq \mathbb{E}_\sigma \left[ \sup_{V \in \mathcal{F}_V} \frac{\gamma}{n} \sum_{i=1}^n \sigma_i V(s'_i) \right] + \mathbb{E}_\sigma \left[ \sup_{V \in \mathcal{F}_V} \frac{1}{n} \sum_{i=1}^n (-\sigma_i) V(s_i) \right]$$

$$= \gamma \cdot \mathbb{E}_\sigma \left[ \sup_{V \in \mathcal{F}_V} \frac{1}{n} \sum_{i=1}^n \sigma_i V(s'_i) \right] + \mathbb{E}_\sigma \left[ \sup_{V \in \mathcal{F}_V} \frac{1}{n} \sum_{i=1}^n \sigma_i V(s_i) \right]$$

In the second term, we used the fact that the distribution of $\{-\sigma_i\}_{i=1}^n$ is the same as $\{\sigma_i\}_{i=1}^n$. The terms are the empirical Rademacher complexities of $\mathcal{F}_V$ over the samples $D_n$.

$$\hat{\mathcal{R}}_{D_n}(\mathcal{A}_{class}) \leq (\gamma + 1)\hat{\mathcal{R}}_{D_n}(\mathcal{F}_V)$$

The function $g_V(A) = \alpha_1 \alpha_2 \mathbb{I}(A \geq 0) f(\frac{A}{\alpha_1})$. Therefore, we can find its Lipschitz constant with respect to $A$ is given by $\alpha_1 \alpha_2 \cdot L(R + 2B)\frac{1}{\alpha_1} = \alpha_2 L(R + 2B)$, where $L$ is the smoothness constant of the function $f$. Then we are ready to apply the Talagrand's Contraction Lemma, $\mathcal{R}_n(g_V \circ \mathcal{A}_{class}) \leq L_{g_V} \mathcal{R}_n(\mathcal{A}_{class})$,

$$\mathcal{R}_n(\mathcal{G}) \leq \frac{\alpha_2 L(R + 2B)}{2} \mathcal{R}_n(\mathcal{A}_{class})$$

$$\leq \frac{(1 + \gamma)\alpha_2 L(R + 2B)}{2} \mathcal{R}_n(\mathcal{F}_V)$$

$$\leq \alpha_2 L(R + 2B) \mathcal{R}_n(\mathcal{F}_V)$$

Finally, we can apply the standard uniform convergence theorem with Rademacher complexity, that the class $g_V \in \mathcal{G}$ of functions with range bounded in $R + (1 + \gamma)B$, with probability at least $1 - \delta/2$:

$$\sup_{V \in \mathcal{F}_V} |\mathbb{E}_{d_\mu}[g_V] - \hat{\mathbb{E}}_{d_\mu}[g_V]| \leq 2\alpha_2 L(R + 2B) \mathcal{R}_n(\mathcal{F}_V) + 2(R + (1 + \gamma)B) \cdot \alpha_2 L(R + 2B) \sqrt{\frac{\ln(4/\delta)}{2n}}$$

For the second term in (19), we can have the following control

$$\hat{\mathbb{E}}_{d_\mu}[g_{V^*}] - \mathbb{E}_{d_\mu}[g_{V^*}] = \alpha_1 \alpha_2 \left( \hat{\mathbb{E}}_{d_\mu}[\mathbb{I}(\hat{A}(s, a) \geq 0) f(\frac{\kappa \hat{A}(s, a)}{\alpha_1})] - \mathbb{E}_{d_\mu}[\mathbb{I}(A_{V^*}(s, a) \geq 0) f(\frac{\kappa A_{V^*}(s, a)}{\alpha_1}))] \right) \quad (21)$$

$$\leq \alpha_1 \alpha_2 \hat{\mathbb{E}}_{d_\mu}[f(\frac{\kappa \hat{A}(s, a)}{\alpha_1})] \quad (22)$$

$$\leq O(\alpha_2 \hat{\mathbb{E}}_{d_\mu}[f(\hat{\Delta}q(s, a))]) \quad (23)$$

$$\leq O(\alpha_2 \epsilon_2) \quad (24)$$

The second inequality holds because $f$ is non-negative. The third equality holds from the lemma A.4 . While the last inequality is directly get from the second level constraint in prime formulation.

**For term (C)** , we control the evaluation error using uniform convergence guarantees.

$$\mathbb{E}_{d_0}[\hat{V}(s_0)] - J_{\hat{\pi}} = \frac{-1}{1-\gamma}\mathbb{E}_{d_{\hat{\pi}}}[A_{\hat{V}}(s,a)] \tag{25}$$

$$\leq \frac{1}{1-\gamma}\left|\mathbb{E}_{d_{\hat{\pi}}}[A_{\hat{V}}(s,a)]\right| \tag{26}$$

$$\leq \frac{1}{1-\gamma}\left(\left|\hat{\mathbb{E}}_{d_{\hat{\pi}}}[A_{\hat{V}}(s,a)]\right| + \left|\hat{\mathbb{E}}_{d_{\hat{\pi}}}[A_{\hat{V}}(s,a)] - \mathbb{E}_{d_{\hat{\pi}}}[A_{\hat{V}}(s,a)]\right|\right) \tag{27}$$

$$\leq O(\frac{\epsilon_1}{1-\gamma}) + \frac{1}{1-\gamma}\left|\hat{\mathbb{E}}_{d_{\hat{\pi}}}[A_{\hat{V}}(s,a)] - \mathbb{E}_{d_{\hat{\pi}}}[A_{\hat{V}}(s,a)]\right| \tag{28}$$

$$\leq O(\frac{\epsilon_1}{1-\gamma}) + \frac{1}{1-\gamma}\left|\hat{\mathbb{E}}_{d_{\mu}}[\hat{\omega}(s,a)A_{\hat{V}}(s,a)] - \mathbb{E}_{d_{\mu}}[\hat{\omega}(s,a)A_{\hat{V}}(s,a)]\right| \tag{29}$$

$$\leq O(\frac{\epsilon_1}{1-\gamma}) + \frac{1}{1-\gamma}\sup_{V\in\mathcal{F}_V}\left|\hat{\mathbb{E}}_{d_{\mu}}[h(A_V)] - \mathbb{E}_{d_{\mu}}[h(A_V)]\right| \tag{30}$$

The third inequality applies the Triangle Inequality. The fourth inequality (28) holds following a similar argument as (A). (29) applies Importance Sampling, defining the weighted advantage composition $h(A) = \hat{\omega}(A) \cdot A$.

To bound the uniform convergence term, we define the class of functions $\mathcal{H} = \{h \circ A_V \mid V \in \mathcal{F}_V\}$, where $h(A) = \max(0, \kappa\alpha_2((f^*)')^{-1}(\frac{\kappa A}{\alpha_1}))A$. Here we used the conjugacy relationship $(f^*)'(y) = (f')^{-1}(y)$ , so the weighting function is $\omega(A) = \max(0, \alpha_2\kappa f'(\frac{\kappa A}{\alpha_1}))$.

We now bound the Lipschitz constant of $h$. Note that $h$ is piecewise differentiable because of the $\max(0,\cdot)$ operator. Let $\mathcal{I} := \{A : \kappa f'(\kappa A/\alpha_1) \leq 0\}$ be the inactive region. On $\mathcal{I}$ we have $\omega(A) = 0$ and hence $h(A) = 0$, so $|h'(A)| = 0$ almost everywhere on $\mathcal{I}$.

On the active region $\mathcal{A} := \{A : \kappa f'(\kappa A/\alpha_1) > 0\}$ we have $\omega(A) = \kappa\alpha_2 f'(\kappa A/\alpha_1)$ and thus, by the product rule,

$$h'(A) = \omega'(A) \cdot A + \omega(A)$$
$$= \left(\kappa\alpha_2 f''\left(\frac{\kappa A}{\alpha_1}\right) \cdot \frac{\kappa}{\alpha_1}\right)A + \kappa\alpha_2 f'\left(\frac{\kappa A}{\alpha_1}\right). \tag{31}$$

Taking absolute values and using $|\kappa| = 1$ gives, for almost all $A \in \mathcal{A}$,

$$|h'(A)| \leq \frac{\alpha_2}{\alpha_1}\left|f''\left(\frac{\kappa A}{\alpha_1}\right)\right| \cdot |A| + \alpha_2\left|f'\left(\frac{\kappa A}{\alpha_1}\right)\right|. \tag{32}$$

Using the assumptions that $f$ is $L$-smooth, and $L(R + 2B)$ globally Lipschitz, since $|A|$ is bounded by $R + 2B$, we obtain

$$|h'(A)| \leq \frac{\alpha_2 L(R+2B)}{\alpha_1} + \alpha_2 L(R+2B) = \alpha_2\frac{\alpha_1+1}{\alpha_1}L(R+2B). \tag{33}$$

Since $\alpha_1 \in [a^{-1}, a]$ implies $\frac{1}{\alpha_1} \leq a$, we conclude that $h$ is globally Lipschitz with

$$L_h \leq \alpha_2(a+1)L(R+2B).$$

Applying Talagrand's Contraction Lemma:

$$\mathcal{R}_n(\mathcal{H}) \leq L_h\mathcal{R}_n(\mathcal{A}_{class}) \leq \alpha_2(a+1)L(R+2B)(1+\gamma)\mathcal{R}_n(\mathcal{F}_V)$$

Finally, applying the standard uniform convergence theorem, with probability at least $1 - \delta/2$:

$$\sup_{V\in\mathcal{F}_V}|\mathbb{E}_{d_{\mu}}[h(A_V)] - \hat{\mathbb{E}}_{d_{\mu}}[h(A_V)]| \leq O\left(\alpha_2 L(R+2B)\mathcal{R}_n(\mathcal{F}_V) + \alpha_2 L(R+2B)^2\sqrt{\frac{\ln(1/\delta)}{n}}\right)$$

**Combine (A), (B), (C).** Taking a union bound over the three high-probability events and combining them yields, with probability at least $1 - \delta$,

$$J_{\pi^*} - J_{\hat{\pi}} \leq \frac{1}{1-\gamma} O\left(\epsilon_1 + \alpha_2 \epsilon_2 + \alpha_2 L (R + 2B)^2 \sqrt{\frac{\ln(1/\delta)}{n}} + \alpha_2 L (R + 2B) \mathcal{R}_n(\mathcal{F}_V)\right)$$

as claimed. □

**Corollary A.6.** *If we instantiate the algorithm by picking $f(x) = \frac{x^2}{2}$, then we can have the following holds with probability at least $1 - \delta$:*

$$J(\pi^*) - J(\hat{\pi}) \leq \frac{1}{1-\gamma} O\left(\epsilon_1 + \sqrt{\epsilon_2} + \frac{(R + 2B)^2}{\sqrt{\epsilon_2}} \sqrt{\frac{\ln(1/\delta)}{n}} + \frac{R + 2B}{\sqrt{\epsilon_2}} \mathcal{R}_n(\mathcal{F}_V)\right)$$

*Proof.* We instantiate the result of the main Theorem using the quadratic loss $f(x) = x^2/2$. First, we observe that this function is 1-smooth, allowing us to set $L = 1$. Substituting these specific values for $L$ into the general bound simplifies the coefficient of the Rademacher complexity term to $O(\alpha_2(R + 2B))$ and the concentration term to $O(\alpha_2(R + 2B)^2)$. The specific form of the corollary is then obtained by solving the dual variable $\alpha_2 = \Theta(\epsilon_2^{-1/2})$ yielding the claimed bound. □

## B. Empirical Connection to Expectile Regression

While we have theoretically established that REG with a squared-loss constraint yields a one-sided regression objective similar to Implicit Q-Learning (IQL), it is crucial to verify this relationship empirically. In IQL, the expectile $\tau \in (0.5, 1)$ acts as a hyperparameter controlling the degree of "optimism" within the dataset support—higher $\tau$ approximates the maximum operator $\max_a Q(s, a)$. In the REG framework, this role is played by the dual variable $\alpha$ (specifically, inversely related to $\epsilon_1$). A smaller $\alpha$ reduces the penalty on the value magnitude, allowing $V(s)$ to push closer to the upper envelope of the sample distribution to minimize the one-sided Bellman error.

To illustrate this behavioral equivalence, we conducted a controlled regression experiment on synthetic two-dimensional data. We generated data $y = 5\sin(\pi x) + \epsilon$, where $x \in [0, 1]$, under three distinct noise profiles: (1) Homogeneous Gaussian noise; (2) Heteroscedastic noise where variance scales with $|y|$; and (3) Boundary-concentrated noise. We then trained a value estimator $V_\psi(x)$ to approximate the conditional distribution of $y$ using both the IQL expectile loss (Eq. 3) and our REG objective (Eq. 9).

Figure 4 presents the results. The first row displays the raw data distribution. The second row shows IQL fits for varying $\tau$, and the third row shows REG fits for varying $\alpha$. We observe a striking symmetry in their behavior. As $\tau \to 1$ in IQL (Yellow curve, $\tau = 0.9$), the learned function approximates the conditional maximum. Similarly, as $\alpha \to 0$ in REG (Yellow curve, $\alpha = 0.01$), the function lifts from the conditional mean to the conditional maximum. This empirically confirms that $\alpha$ serves as the natural dual counterpart to the expectile $\tau$. In the middle column (Heteroscedastic noise), both methods successfully adapt to the changing variance, maintaining a tight bound on the upper envelope of the data without diverging. In the third column, where noise is concentrated at the boundaries, both methods exhibit stable behavior, avoiding the "chasing outliers" phenomenon often seen in unconstrained maximization. This experiment reinforces our theoretical claim: REG recovers the desirable "in-sample maximization" property of IQL through a principled robust optimization formulation, rather than a heuristic choice of loss function.

## C. Experimental Details

We base our implementation of REG on the official JAX implementation of IQL. We modified the network architecture and apply the Double-Q trick. We also apply the same data preprocessing which is described in their appendix.

**D4RL datasets details.** The "medium" dataset is generated by first training a policy online using Soft Actor-Critic, early-stopping the training, and collecting 1M samples from this partially-trained policy. The "medium-replay" dataset consists of recording all samples in the replay buffer observed during training until the policy reaches the "medium" level of performance. The "medium-expert" dataset is built by mixing equal amounts of expert demonstrations and suboptimal data, generated via a partially trained policy or by unrolling a uniform-at-random policy.

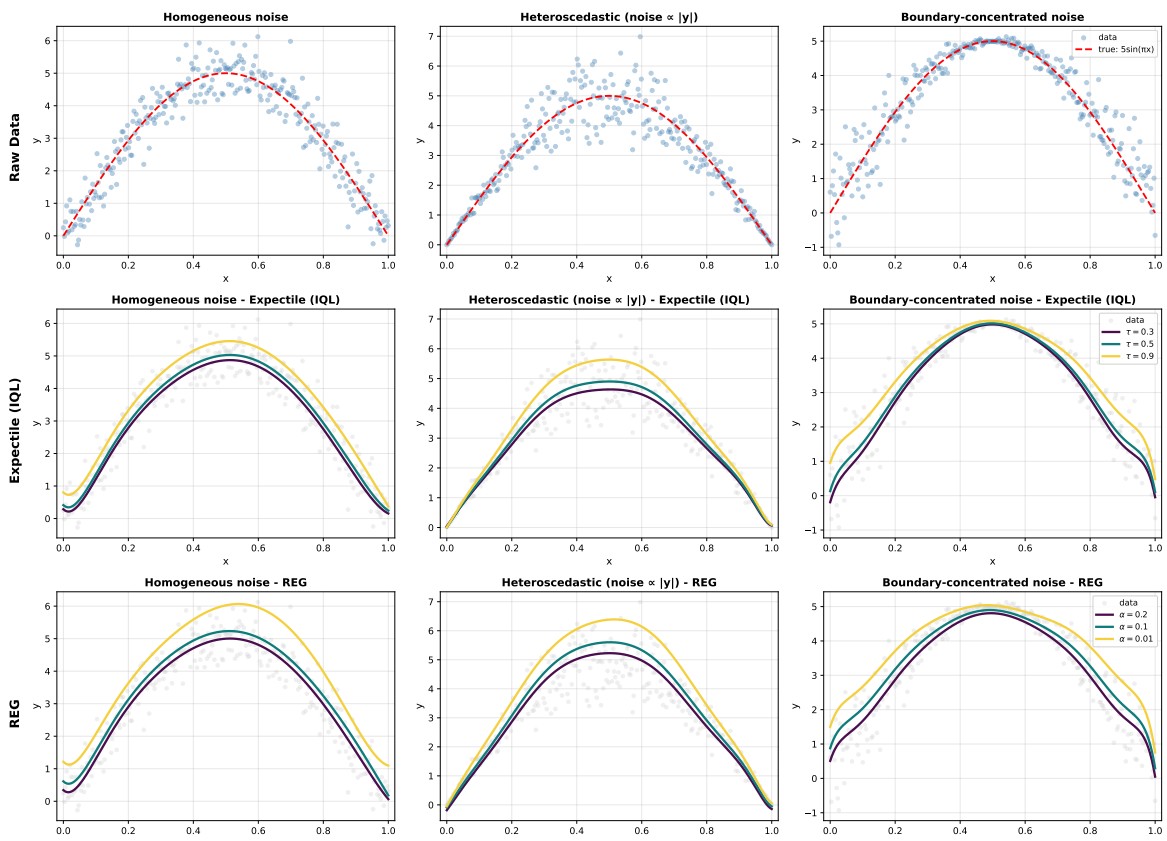

*Figure 4.* First Row: **Synthetic two-dimensional data**. From left to right 1) Homogeneous noise 2) Noise proportion to $|y|$ 3) Noise at the boundary. Second Row: **Expectile Regression**. Fits using expectile loss with three values ($\tau = 0.3, 0.5, 0.9$). Third Row: **REG Regression**. Fits using REG loss with three values ($\alpha = 0.2, 0.1, 0.01$).

**D4RL experimental details.** For all tasks, we conducted our algorithm for $10^6$ steps and reported the final performance. In MuJoCo locomotion tasks, we computed the average returns over 10 evaluations every 10k steps, across 5 different seeds. For AntMaze tasks, we calculated the average over 50 evaluations every 20k steps, also across 5 seeds. Followed by IQL, we standardize the rewards by dividing the difference in returns of the best and worst trajectories in MuJoCo and Kitchen tasks, we subtract 1 to rewards in AntMaze tasks.

For the MuJoCo environment, we adapt a 2-layer MLP with ReLU activation, and we use 256 hidden units for all networks. We choose the Adam optimizer (Kingma & Ba, 2015) with a learning rate of $2 \times 10^{-4}$ and batch size of 512 for both $V, Q$, and $\pi$ in all tasks. We use a target network with soft update weight $5 \times 10^{-3}$ for V. For the AntMaze environment, we adapt a 4-layer MLP with swish activation, and we use [256, 256, 128, 64] hidden units for all the networks. We chose Adam optimizer with a learning rate of $2 \times 10^{-4}$ and batch size of 512 for both $V, Q$, and $\pi$ in all tasks. We use a target network with soft update weight $5 \times 10^{-3}$ for V. The task-specific hyperparameters are listed in Table 2.

| Dataset | $\alpha$ | $\lambda$ |
|---|---|---|
| halfcheetah-m | 0.2 | 0.3 |
| hopper-m | 0.8 | 0.4 |
| walker2d-m | 0.8 | 0.5 |
| halfcheetah-m-r | 0.2 | 0.6 |
| hopper-m-r | 0.8 | 0.1 |
| walker2d-m-r | 0.8 | 0.4 |
| halfcheetah-m-e | 0.2 | 0.0 |
| hopper-m-e | 0.8 | 0.0 |
| walker2d-m-e | 0.8 | 0.6 |
| antmaze-umaze | 0.2 | 0.4 |
| antmaze-umaze-diverse | 0.5 | 0.1 |
| antmaze-medium-play | 0.2 | 0.4 |
| antmaze-medium-diverse | 0.2 | 0.4 |
| antmaze-large-play | 0.2 | 0.4 |
| antmaze-large-diverse | 0.2 | 0.4 |

*Table 2.* Hyperparameters configuration: $\alpha$ controls the confidence in the offline dataset, $\lambda$ controls the strength of the orthogonal policy gradient.

*Table 3.* Sensitivity analysis for the REG hyperparameter $\alpha$. Scores are normalized D4RL scores, averaged over 5 seeds. The hyperparameter $\lambda$ is held fixed for each environment according to our main experiments. The best performing $\alpha$ for each task is highlighted.

| Dataset | $\alpha = 0.2$ | $\alpha = 0.4$ | $\alpha = 0.6$ | $\alpha = 0.8$ | $\alpha = 1.0$ |
|---|---|---|---|---|---|
| halfcheetah-m | 49.6 ± 0.4 | 49.4 ± 0.2 | 48.9 ± 0.3 | 48.7 ± 0.0 | 48.6 ± 0.2 |
| hopper-m | 56.9 ± 2.0 | 61.9 ± 7.4 | 64.1 ± 3.5 | 74.4 ± 4.0 | 70.2 ± 3.6 |
| walker2d-m | 31.5 ± 8.5 | 46.4 ± 3.7 | 60.5 ± 11.9 | 80.3 ± 7.6 | 82.4 ± 3.4 |
| halfcheetah-m-r | 44.5 ± 0.6 | 43.9 ± 2.0 | 43.8 ± 2.3 | 42.9 ± 1.8 | 44.3 ± 0.7 |
| hopper-m-r | 6.8 ± 2.3 | 81.3 ± 21.9 | 100.9 ± 0.5 | 101.3 ± 0.9 | 95.2 ± 11.3 |
| walker2d-m-r | 5.2 ± 0.5 | 6.7 ± 1.5 | 12.4 ± 2.5 | 78.6 ± 4.3 | 66.8 ± 6.5 |
| halfcheetah-m-e | 94.1 ± 0.6 | 93.4 ± 0.9 | 84.8 ± 4.6 | 85.4 ± 3.9 | 81.2 ± 2.4 |
| hopper-m-e | 108.7 ± 1.3 | 102.0 ± 6.3 | 107.2 ± 1.8 | 110.6 ± 0.9 | 109.4 ± 1.1 |
| walker2d-m-e | 111.8 ± 0.2 | 111.6 ± 0.3 | 111.2 ± 0.4 | 110.9 ± 0.2 | 110.8 ± 0.3 |

*Table 4.* Sensitivity analysis for the OPG hyperparameter $\lambda$. Scores are normalized D4RL scores, averaged over 5 seeds. The hyperparameter $\alpha$ is held fixed for each environment according to our main experiments. The best performing $\lambda$ for each task is highlighted.

| Dataset | $\lambda = 0.0$ | $\lambda = 0.1$ | $\lambda = 0.2$ | $\lambda = 0.3$ | $\lambda = 0.4$ | $\lambda = 0.5$ | $\lambda = 0.6$ |
|---|---|---|---|---|---|---|---|
| halfcheetah-m | 49.6 ± 0.4 | 50.9 ± 0.2 | 51.3 ± 0.3 | 53.6 ± 0.5 | 51.9 ± 0.5 | 52.2 ± 0.8 | 38.9 ± 12.3 |
| hopper-m | 74.4 ± 4.0 | 77.1 ± 7.5 | 82.6 ± 4.5 | 85.6 ± 6.5 | 91.9 ± 6.7 | 91.4 ± 9.8 | 81.4 ± 14.9 |
| walker2d-m | 80.3 ± 7.6 | 85.7 ± 0.2 | 85.3 ± 1.0 | 86.1 ± 0.8 | 86.5 ± 0.4 | 87.9 ± 1.7 | 86.9 ± 0.5 |
| halfcheetah-m-r | 44.5 ± 0.6 | 47.2 ± 0.5 | 48.5 ± 0.4 | 49.4 ± 0.4 | 49.6 ± 0.5 | 49.5 ± 0.2 | 50.0 ± 0.2 |
| hopper-m-r | 101.3 ± 0.9 | 101.4 ± 0.5 | 92.8 ± 3.6 | 90.2 ± 12.2 | 96.7 ± 5.4 | 100.5 ± 0.7 | 100.2 ± 2.6 |
| walker2d-m-r | 78.6 ± 4.3 | 73.9 ± 5.9 | 72.8 ± 7.5 | 82.1 ± 8.1 | 87.3 ± 4.3 | 81.2 ± 5.7 | 83.8 ± 6.6 |
| halfcheetah-m-e | 94.1 ± 0.6 | 84.3 ± 5.9 | 53.5 ± 2.4 | 51.0 ± 5.9 | 47.9 ± 10.7 | 43.5 ± 1.4 | 43.0 ± 2.8 |
| hopper-m-e | 110.6 ± 0.9 | 93.1 ± 13.5 | 29.0 ± 7.0 | 31.3 ± 2.1 | 32.6 ± 5.2 | 28.1 ± 5.1 | 12.1 ± 5.9 |
| walker2d-m-e | 110.9 ± 0.2 | 112.4 ± 0.4 | 112.0 ± 0.6 | 112.9 ± 0.0 | 113.2 ± 0.3 | 113.0 ± 0.3 | 113.5 ± 0.8 |

