# OpenReview forum: "REG: In-Sample RL via Regularizing the Evaluation Gap"
_ICML.cc/2026/Conference — ICML 2026 regular_

### Official Review · Reviewer_TKg9 · 2026-03-10

**Soundness:** 3
**Presentation:** 2
**Significance:** 3
**Originality:** 3
**Overall Recommendation:** 4
**Confidence:** 3

**Summary:**

This paper introduces REG to address the distribution shift issue in offline RL via bridging IQL and explicit conservative value estimation. Specifically, REG presents the robust optimization formulation for offline RL with theoretical analysis on trade-offs among conservativeness, robustness, uncertainty, and statistical error. Experimental results on D4RL demonstrate the advanced performance of the proposed method.

**Compliance With Llm Reviewing Policy:**

Affirmed.

**Final Justification:**

My concerns have been adequately addressed.

**Key Questions For Authors:**

1. Can the proposed method be extended to diffusion policy?
2. Some baselines (TD3+BC[1] and TD7[2]) should be considered.


[1] A Minimalist Approach to Offline Reinforcement Learning

[2] For SALE: State-Action Representation Learning for Deep Reinforcement Learning

**Limitations:**

1. Gaussian policy may struggle to address multimodal behavior distribution.
2. The performance is sensitive to the choice of specific parameters ($\alpha$ and $\lambda$).

**Strengths And Weaknesses:**

## Strengths
1. Presents a robust optimization framework for offline RL.
2. Theoretical analysis on trade-offs among conservativeness, robustness, uncertainty, and statistical error.


## Weaknesses
1. The performance is likely sensitive to the choice of $\alpha$ of conservatism and $\lambda$ of OPG strength.
2. The Gaussian policy may struggle to address multimodal behavior distribution.

---

> ### Author Rebuttal · Authors · 2026-03-31
>
> We sincerely thank you for your time and valuable feedback. We appreciate your constructive suggestions regarding hyperparameter sensitivity, multi-modal behavior, and additional baselines, which help strengthen the paper.
>
> >  The performance is likely sensitive to the choice of $\alpha$ of conservatism and $\lambda$ of OPG strength.
>
> We agree that hyperparameter tuning is important in offline RL. However, this is a common challenge across many RL algorithms. Methods like IQL and ReBRAC also rely on task-specific parameters for value learning and policy extraction. For instance, while IQL uses a constant hyperparameter for all locomotion datasets, CORL [1] experiments show that different tasks, such as Hopper-medium, require fine-tuning these hyperparameters for optimal performance.
>
> In our locomotion experiments, we found that the conservativeness parameter is relatively robust across tasks, and as shown in Table 1, we report the performance under a fully tuned setting. On the other hand, the exploratory strength parameter is more sensitive to the environment. However, even when limiting ourselves to two values for each parameter, the performance does not degrade significantly. We present these results below, which highlight the trade-offs between different hyperparameter choices:
>
> |Dataset|REG (Fully Tuned)|REG (α=0.8,λ=0.0)|REG (α=0.8,λ∈{0.0,0.4})|
> |-|-|-|-|
> |halfcheetah-m|53.6±0.5|48.7±0.0|51.6±0.5|
> |hopper-m|91.9±6.7|74.4±4.0|91.9±6.7|
> |walker2d-m|87.9±1.7|80.3±7.6|86.5±0.4|
> |halfcheetah-m-r|50.0±0.2|42.9±1.8|46.9±0.9|
> |hopper-m-r|101.4±0.5|101.3±0.9|96.7±5.4|
> |walker2d-m-r|87.3±4.3|78.6±4.3|87.3±4.3|
> |halfcheetah-m-e|94.1±0.6|85.4±3.9|94.1±0.6|
> |hopper-m-e|110.6±0.9|110.6±0.9|110.6±0.9|
> |walker2d-m-e|113.5±0.8|110.9±0.2|110.9±0.2|
> |**Average**|**87.9±2.8**|**81.5±3.5**|**86.2±3.3**|
>
> > The Gaussian policy may struggle to address multimodal behavior distribution.
>
> We acknowledge that Gaussian policies have limitations in handling multimodal behavior distributions. Our choice of a Gaussian policy was primarily to validate the effectiveness of the REG framework. We consider it a strength that REG performs so well despite using this cheaper policy class. Diffusion-based policies represent a promising direction for further improvement, and we plan to explore their integration with REG in future work.
>
> > Can the proposed method be extended to diffusion policy?
>
> Yes, we can extend the framework to diffusion policy (Same as the response to Reviewer mN9j), the reason is because maximizing the weighted log-likelihood is equivalent to minimizing the score matching loss.
>
> As detailed in our response to Reviewer mN9j, we swapped our default Gaussian actor for a diffusion model while retaining REG's robust value learning objective. The results show that REG effectively supports and enhances downstream diffusion policy learning, achieving highly competitive performance. This demonstrates that the core REG framework readily accommodates and fundamentally benefits highly expressive policy classes out-of-the-box.
>
> While our specific Orthogonal Policy Gradient (OPG) component is designed to efficiently overcome the mode covering issue of mode covering from minimizing the forward KL, and adapting it to the score-matching mechanisms of diffusion models remains an interesting direction for future work. The core REG critic framework is compatible with diffusion models.
> |Dataset|IQL+Gaussian|REG+Gaussian|IQL+Diffusion(IDQL-1)|REG+Diffusion(REGD-1)|
> |-|-|-|-|-|
> |halfcheetah-m|47.4±0.2|49.6±0.4|49.7|**49.5**|
> |hopper-m|66.3±5.7|**74.4±4.0**|63.1|69.7|
> |walker2d-m|72.5±8.7|80.3±7.6|80.2|**85.6**|
> |halfcheetah-m-r|44.2±1.2|44.5±0.6|45.1|**46.7**|
> |hopper-m-r|95.2±8.6|**101.3±0.9**|82.4|83.1|
> |walker2d-m-r|76.1±7.3|**78.6±4.3**|79.8|77.7|
> |halfcheetah-m-e|86.7±5.3|94.1±0.6|94.4|**106.2**|
> |hopper-m-e|101.5±7.3|**110.6±0.9**|105.3|110.1|
> |walker2d-m-e|110.6±1.0|110.9±0.2|**111.6**|111.2|
> > Some baselines (TD3+BC[1] and TD7[2]) should be considered.
>
> Thank you for the suggestion. We have cited TD3+BC in the literature but chose not to include it in the baseline comparison as its performance is lower than IQL and several other baselines. TD7, however, is a relevant baseline and yields competitive results. We include the following table comparing REG, TD3+BC, and TD7:
> ||REG|TD3+BC|TD7|
> |-|-|-|-|
> |Halfcheetah-m|53.6±0.5|48.1±0.1|58.0±0.4|
> |Hopper-m|91.9±6.7|59.1±3.0|76.1±5.1|
> |Walker2d-m|87.9±1.7|84.3±0.8|91.1±7.8|
> |Halfcheetah-m-r|50.0±0.2|44.6±0.4|53.8±0.8|
> |Hopper-m-r|101.4±0.5|52.0±10.6|91.1±8.0|
> |Walker2d-m-r|87.3±4.3|81.0±3.4|89.7±4.7|
> |Halfcheetah-m-e|94.1±0.6|93.7±0.9|104.6±1.6|
> |Hopper-m-e|110.6±0.9|98.1±10.7|108.2±4.8|
> |Walker2d-m-e|113.5±0.8|110.5±0.4|111.8±0.6|
> |Average|**87.8±2.8**|74.6±5.23|**87.2±4.7**|
>
> [1] Tarasov, D., Nikulin, A., Akimov, D., Kurenkov, V., and Kolesnikov, S. CORL: Research-oriented deep offline reinforcement learning library. In *3rd Offline RL Workshop: Offline RL as a ''Launchpad''*, 2022.

---

> > ### Author Rebuttal · Reviewer_TKg9 · 2026-04-03
> >
> > Thank you for the detailed reply. All my concerns have been adequately addressed.

---

> > > ### Author Response · Authors · 2026-04-03
> > >
> > > We sincerely appreciate your active engagement during the discussion phase and thank you for raising your score. We are glad that our ablation studies and clarifications resolved your questions. Thank you again for your valuable time and for your efforts in helping us strengthen our work.

---

### Official Review · Reviewer_4UR2 · 2026-03-11

**Soundness:** 3
**Presentation:** 3
**Significance:** 3
**Originality:** 4
**Overall Recommendation:** 5
**Confidence:** 2

**Summary:**

The paper introduces REG, a method to approach the problem out-of-distribution estimation in offline RL. The authors propose to reframe in-sample learning as a constraint optimisation problem on the learnable Q-function. To overcome the intractability of this hard-constrained optimisation problem, Langrangian Duality is employed to translate the problem into a tractable, soft-constrained problem. This soft-constrained optimisation problem serves as the foundation for their practical algorithm, that demonstrates strong empirical results.

**Compliance With Llm Reviewing Policy:**

Affirmed.

**Key Questions For Authors:**

- Both the theoretical analysis and IQL (on which the implementation is based) suggest first training V and Q, and only afterwards perform a policy extraction phase. However, the authors choose to intertwine these two steps. Some further details on this choice could be a valuable addition.
- In relation to the second weakness above, the orthogonal policy gradient appears to be a technique that could also be applied to other methods, such as IQL, to potentially obtain performance improvements. Therefore, to better evaluate the core method itself, a comparison without the use of said orthogonal gradient could be a useful addition.
- The proposed algorithm, like IQL, has the appeal of being conceptually easy to understand. I wonder whether this conceptual simplicity (in a positive sense) also translates into faster convergence in terms of runtime and number of required iterations.

**Limitations:**

yes

**Strengths And Weaknesses:**

Strengths:
- Very clear motivation and derivation, with a coherent storyline that provides a useful angle on existing methods.
- Fair comparison: The results in Table 1 align with the values reported in the respective papers and can therefore be expected to provide a competitive baseline.
- While I have limited experience in the field, the paper appears to provide an interesting angle for creating a new set of methods by controlling the Bellman error. Although the results do not outperform state-of-the-art methods (see weaknesses), I believe that introducing this perspective is valuable in itself.

Weaknesses:
- While the work provides a novel perspective on regularization in offline RL, even with the additional design choices (such as orthogonal policy gradient), the method does not outperform state-of-the-art approaches, including those that use Gaussian policies (e.g., ReBRAC).
- The method introduces two new hyperparameters: $\lambda$ and $\alpha$. As table 2 suggests, these have to be tuned per environment to reach the reported performance, introducing additional undesired complexity.
- Minor:
    - Running title: “Submission and Formatting Instructions for ICML 2026”
    - No available code




Overall, this reads like a solid paper: very well grounded, structured, scoped, and executed, with convincing results both theoretically and practically. However, I have limited experience in the area of offline RL and would therefore like to ask the AC to incorporate my review accordingly.

---

> ### Author Rebuttal · Authors · 2026-03-31
>
> We sincerely thank you for your careful reading of our paper and your positive, constructive feedback. We greatly appreciate the opportunity to clarify our approach and address the questions you raised. Below, we respond to your key points:
>
> >While the work provides a novel perspective on regularization in offline RL, even with the additional design choices (such as orthogonal policy gradient), the method does not outperform state-of-the-art approaches, including those that use Gaussian policies (e.g., ReBRAC)
>
> We appreciate your comment. REG’s performance is competitive with state-of-the-art methods, but we acknowledge that it does not always outperform the most advanced offline RL approaches. Our method’s novelty lies in its robust optimization framework, which connects implicit value learning methods like IQL to dual solutions of constrained problems. REG’s strength is in the simplicity and efficiency of its Gaussian-policy architecture, which avoids the computational burden of diffusion models.
>
> > The method introduces two new hyperparameters: $\lambda$ and $\alpha$. As table 2 suggests, these have to be tuned per environment to reach the reported performance, introducing additional undesired complexity.
>
> We acknowledge your concern about hyperparameter tuning. However, this issue is common in RL algorithms. Methods like IQL and ReBRAC also require task-specific hyperparameters for value learning and policy extraction. REG’s hyperparameters (α and λ) show reasonable robustness across domains, and we demonstrate this in Tables 2 and 3. While some tuning is needed, performance is relatively stable across tasks, and we report results under fully tuned settings. Additionally, even when limiting to just two values for each parameter, performance does not degrade significantly (first Table in the response to reviewer mN9j).
>
> > Running title: “Submission and Formatting Instructions for ICML 2026”
>
> Thank you for pointing this out! We will correct it in the final submission.
>
> > No available code
>
> The code will be released upon acceptance.
>
> > Both the theoretical analysis and IQL (on which the implementation is based) suggest first training V and Q, and only afterwards perform a policy extraction phase. However, the authors choose to intertwine these two steps. Some further details on this choice could be a valuable addition.
>
> You are correct in noting that offline RL methods often separate value and policy learning. However, our choice to intertwine these steps is driven by practical considerations. REG is not an actor–critic algorithm, and interleaved value and policy updates are not necessary here as they are in online policy-gradient algorithms. By updating the value functions V and Q and the policy simultaneously, we better utilize hardware to reduce training wall-time as we don't need to wait for value convergence before starting policy learning. In practice, we observed that early instability in value learning led to some chattering in policy performance, which stabilized after value learning converged. This design choice aligns with the efficient training practices of methods like IQL and ReBRAC.
>
> > In relation to the second weakness above, the orthogonal policy gradient appears to be a technique that could also be applied to other methods, such as IQL, to potentially obtain performance improvements. Therefore, to better evaluate the core method itself, a comparison without the use of said orthogonal gradient could be a useful addition.
>
> We appreciate this suggestion. We have included a comparison in Figure 2. And we will provide a more detailed comparison table in the camera-ready version to highlight the effect of Orthogonal Policy Gradient (OPG) relative to the baseline methods.
>
> > The proposed algorithm, like IQL, has the appeal of being conceptually easy to understand. I wonder whether this conceptual simplicity (in a positive sense) also translates into faster convergence in terms of runtime and number of required iterations.
>
> Thank you for bring up this point. We agree that the conceptual simplicity of REG is a strength. In terms of runtime, REG is similar to IQL, with approximately 20 minutes for 1M training steps on an H20. However, convergence speed is highly dependent on the learning rate and environment.

---

> > ### Author Rebuttal · Reviewer_4UR2 · 2026-04-01
> >
> > Thank you for the detailed reply. My concerns have been adequately addressed.

---

> > > ### Author Response · Authors · 2026-04-07
> > >
> > > Thank you very much for your thoughtful review and for confirming that our rebuttal addressed your concerns. We truly appreciate the time you took to engage with our responses and for your constructive feedback, which has helped us improve the quality and clarity of our work

---

### Official Review · Reviewer_mN9j · 2026-03-12

**Soundness:** 3
**Presentation:** 2
**Significance:** 3
**Originality:** 3
**Overall Recommendation:** 4
**Confidence:** 4

**Summary:**

This paper introduces Regularized Evaluation Gap (REG), a novel offline reinforcement learning framework that provides a principled robust optimization foundation for implicit value learning. By formulating policy evaluation as an optimization problem over an ambiguity set of data-consistent critics, the authors demonstrate that the asymmetric objectives used in popular methods like IQL are approximate dual solutions to this robust formulation rather than mere heuristics. To effectively extract a policy from this learned value function, the authors propose a practical Orthogonal Policy Gradient (OPG) update, which adaptively balances conservative, in-sample weighted behavior cloning with aggressive, mode-seeking exploration by projecting the policy gradient onto the subspace orthogonal to the conservative gradient. Extensive empirical evaluations on the D4RL benchmark suite show that REG matches the state-of-the-art performance of both Gaussian and complex diffusion-based methods while maintaining a significantly simpler and more computationally efficient architecture.

**Compliance With Llm Reviewing Policy:**

Affirmed.

**Final Justification:**

The rebuttal addressed my concerns about the hyperparameter sensitivity and contribution.

**Key Questions For Authors:**

1. While REG shows strong performance, it addresses value overestimation (Problem A), whereas Diffusion models focus on policy expressivity (Problem B). Table 1 demonstrates that REG is a robust algorithm and comparable to a few diffusion-based baselines; it is insufficient to show that critic evaluation is the "core issue" (as value estimation and policy expressiveness are generally considered as two separate problems).  Please clarify the evidence of the contribution of REG and its connection to policy expressiveness more clearly.

2. Have the authors considered an ablation using REG with a Diffusion policy? This would clarify whether the observed gains are strictly due to the value objective or if the Gaussian architecture + OPG is simply an effective, yet distinct, optimization strategy for specific D4RL modes.

As the major message of the paper is somewhat confusing, I would temporarily recommend reject until the confusion has been clarified.

**Limitations:**

Yes

**Strengths And Weaknesses:**

### Strengths
1. The paper provides a principled robust optimization framework that theoretically connects IQL's heuristic objectives to a dual solution of a constrained problem.
2. Claims are supported by a finite-sample PAC regret analysis.
3. The method is rigorously tested on the D4RL benchmark, demonstrating state-of-the-art results across locomotion and navigation tasks.

### Weakness:
1. The algorithm's performance relies on hyperparameters like $\alpha$ (conservativeness) and $\lambda$ (exploratory strength), which require manual tuning for different tasks.
2. Conceptual Conflation of Two Distinct Problems: The paper treats value estimation and policy expressivity as if they are on the same axis of "the core issue". However, these are two separate problems that
3. Empirical Ambiguity: The experiment design does not isolate the value estimation and policy expressiveness problem well.

---

> ### Author Rebuttal · Authors · 2026-03-31
>
> We sincerely thank you for your detailed review and insightful questions. We truly appreciate the opportunity to clarify these key aspects of our work. Below, we address your comments regarding hyperparameter sensitivity and the distinction between value estimation and policy expressivity, and present the requested ablation with diffusion policies.
>
> > Hyperparameter sensitivity
>
> We agree with your observation that hyperparameter tuning is necessary, but hyperparameter tuning is standard in offline RL; e.g., CORL [1] shows IQL requires per-task tuning for optimal results. In REG, α is highly robust across tasks, while λ is more environment-sensitive. However, when limiting ourselves to two values for λ, the performance does not degrade significantly. We present these results below, which highlight the trade-offs between different hyperparameter choices.
>
> |Dataset|REG (Fully Tuned)|REG (α=0.8,λ=0.0)|REG (α=0.8,λ∈{0.0,0.4})|
> |-|-|-|-|
> |halfcheetah-m|53.6±0.5|48.7±0.0|51.6±0.5|
> |hopper-m|91.9±6.7|74.4±4.0|91.9±6.7|
> |walker2d-m|87.9±1.7|80.3±7.6|86.5±0.4|
> |halfcheetah-m-r|50.0±0.2|42.9±1.8|46.9±0.9|
> |hopper-m-r|101.4±0.5|101.3±0.9|96.7±5.4|
> |walker2d-m-r|87.3±4.3|78.6±4.3|87.3±4.3|
> |halfcheetah-m-e|94.1±0.6|85.4±3.9|94.1±0.6|
> |hopper-m-e|110.6±0.9|110.6±0.9|110.6±0.9|
> |walker2d-m-e|113.5±0.8|110.9±0.2|110.9±0.2|
> |**Average**|**87.9±2.8**|**81.5±3.5**|**86.2±3.3**|
>
> > Weakness 2 and Value estimation (Prob. A) vs. Policy expressivity (Prob. B):
>
> We completely agree with your assessment: Prob. A and B are distinct challenges. To be clear, our core claim is not that REG solves Prob. B. Our framework explicitly addresses Prob. A by providing a robust value learning objective that mitigates overestimation. We do not claim that a better advantage estimate can compensate for a lack of policy expressiveness in highly multimodal environments. Instead, our results demonstrate that in many standard continuous control benchmarks, Prob. B is not yet the primary limiting factor. In short, REG provides significant benefits specifically in domains where extreme policy expressivity is not the bottleneck. However, because our value estimation is independent of policy expressiveness, REG's robust critic can readily be combined with highly expressive policies when Prob. B does become a limiting factor. This can be demonstrated by our diffusion extension below.
>
> > Have the authors considered an ablation using REG with a Diffusion policy? And Weakness 3
>
> We agree that this is the cleanest way to separate the contributions of value objective from Gaussian+OPG optimization. We ran an ablation study where we fix REG’s value learning and swap only the actor class. Specifically, we will compare the following configurations:
>
> - Fully tuned: IQL + Gaussian, REG + Gaussian
> - Single hyperparameter: IQL + Diffusion (IDQL-1), REG + Diffusion (REGD-1)
>
> In addition, we conducted a comparison with IDQL, a diffusion-based adaptation of IQL. This experiment serves to validate the effectiveness of the value function induced from our robust learning framework in supporting downstream diffusion policy learning. Although this adaptation does not strictly follow the REG approach for policy extraction, we include it to demonstrate that the value learning objective induced by REG can enhance policy learning for both Gaussian and Diffusion policy classes. For fairness, we will use the same neural architecture and learning parameters as IDQL, adjusting only the conservativeness parameter $\alpha$ in REG. To ensure consistency, we will use a constant hyperparameter ($\alpha=0.8$) for all locomotion tasks. This experiment reveals that the value learning objective induced by the REG framework can benifit the downstream policy learning for both Gaussian and diffusion policy classes.
>
> OPG is a meaningful add-on that help overcome the mode-covering issues due to policy extraction by minimizing the forward KL $\arg\min_\theta E_s [KL(\pi^* \| \pi_\theta)] $. While policy gradient directly updates the policy along the direction that maximizes the expected cumulative rewards, as evaluated by the learned critic, that is, $\nabla_\theta E_{s\sim D,a\sim\pi_\theta(\cdot|s)}[Q(s,a)]$. We acknowledge that adapting OPG to diffusion models is non-trivial, and thus we plan to leave REG + OPG + Diffusion for future work.
> |Dataset|IQL+Gaussian|REG+Gaussian|IQL+Diffusion(IDQL-1)|REG+Diffusion(REGD-1)|
> |-|-|-|-|-|
> |halfcheetah-m|47.4±0.2|49.6±0.4|**49.7**|49.5|
> |hopper-m|66.3±5.7|**74.4±4.0**|63.1|69.7|
> |walker2d-m|72.5±8.7|80.3±7.6|80.2|**85.6**|
> |halfcheetah-m-r|44.2±1.2|44.5±0.6|45.1|**46.7**|
> |hopper-m-r|95.2±8.6|**101.3±0.9**|82.4|83.1|
> |walker2d-m-r|76.1±7.3|78.6±4.3|**79.8**|77.7|
> |halfcheetah-m-e|86.7±5.3|94.1±0.6|94.4|**106.2**|
> |hopper-m-e|101.5±7.3|**110.6±0.9**|105.3|110.1|
> |walker2d-m-e|110.6±1.0|110.9±0.2|**111.6**|111.2|
>
> [1] Tarasov, D., Nikulin, A., Akimov, D., Kurenkov, V., and Kolesnikov, S. CORL: Research-oriented deep offline reinforcement learning library.

---

> > ### Author Rebuttal · Reviewer_mN9j · 2026-04-01
> >
> > Thank the authors for the detailed reply. all my concerns have been addressed. I have updated my score accordingly.

---

> > > ### Author Response · Authors · 2026-04-03
> > >
> > > Thank you very much for your time, your constructive feedback throughout the review process, and for updating your score. We are very glad that our rebuttal addressed your concerns. We will ensure that all the clarifications and additional ablations discussed during this review phase are carefully incorporated into the final version of the paper.

---

### Official Review · Reviewer_64Y5 · 2026-03-13

**Soundness:** 3
**Presentation:** 3
**Significance:** 3
**Originality:** 3
**Overall Recommendation:** 4
**Confidence:** 3

**Summary:**

This paper formulates the policy evaluation in offline reinforcement learning as a robust optimization problem over a set of possible critics and proposes a dual formulation to avoid solving the intractable primal problem. Then the authors show that the optimal policy can be extracted using weighted BC or orthogonal policy gradients based on the occupancy measure ratio derived in Theorem 4.2. The effectiveness of the proposed algorithm is demonstrated by both theoretical analysis of the sample complexity and experiments on the D4RL benchmarks.

**Compliance With Llm Reviewing Policy:**

Affirmed.

**Final Justification:**

My concerns have been adequately addressed in the rebuttal.

**Key Questions For Authors:**

1. What is the effect of different $f$ function in policy learning?
2. Are there any empirical results on the policy learning performance with different $f$ function?
2. What is the intuition behind the policy gradient $g_{pg}$? The $g_{wbc}$ is straightforward as it is essentially doing MLE, but the $g_{pg}$ is not. In the usual policy gradient, the expectation is taken over $\pi_{old}(a|s)$ and the weighting is the ratio $\frac{\pi_\theta(a|s)}{\pi_{old}(a|s)}$, but in $g_{pg}$ , the expectation is taken over $\pi_{\theta}(a|s)$, and the ratio is between the occupancy measure.

**Limitations:**

The limitations of this work should be discussed more thoroughly.

**Strengths And Weaknesses:**

Strengths

1. The proposed robust optimization formulation and its dual form are novel to the domain of offline RL to the best of my knowledge, and gives new insights to the offline RL field.
2. The experiment results are solid and the theoretical analysis is rigorous.
3. The paper is well-organized and clearly written.

Weaknesses

The authors discussed a general form of $f$ function in Theorem 4.2, but only the special case of $f(x)=x^2/2$ was evaluated in the experiments. It is not clear how different choices of $f$ will affect the policy performance.

---

> ### Author Rebuttal · Authors · 2026-03-30
>
> We sincerely thank you for your time and constructive feedback. We appreciate your thoughtful questions, and we agree that empirically validating alternative choices of $f$ strengthens the generality of Theorem 4.2. Below we clarify (i) why we instantiated $f$ with the squared loss in the submission, (ii) how f affects policy learning in our framework, and (iii) provide an additional ablation with an alternative $f$.
>
> > The authors discussed a general form of f function in Theorem 4.2, but only the special case of f was evaluated in the experiments. It is not clear how different choices of f will affect the policy performance.
>
> Theorem 4.2 is stated for a general differentiable convex $f$ (with $f(0)=0$), and in the submission we instantiate a canonical choice $f(x)=x^2/2$. First, we want to point out that our choice of square loss is the most natural and common choice for regression tasks, and it is arguably the optimal choice if the underlying error distribution is Gaussian. Additionally, squared loss yields a simple closed-form weight map, stable optimization—making it ideal as the primary choice for study. That said, we acknowledge that other choices of loss function can be valuable as alternative implicit assumptions that a practitioner could make about the underlying error distribution.
>
> > What is the effect of different $f$ function in policy learning?
>
> In our framework, $f$ does not change the overall algorithmic structure, but it does change the advantage weight mapping that drives policy learning. Theorem 4.2 shows that the implied optimal weighting function is $\omega(s,a) \propto \max(0,\kappa f'(\kappa A(s,a)/\alpha))$. Therefore, different choices of f primarily change:
>
> 1.  how aggressively high-advantage actions are up-weighted (tail behavior).
> 2.  how sensitive the update is to large advantages which potentially can be outliers, via the shape of $f'$.
>
> In Sec. 4.4 we instantiate $f(x)=x^2/2$ (Gaussian Bellman-error assumption), yielding $\omega^* \propto \max(0, A)$, i.e., linear weighting; other convex f choices yield alternative weighting schedules. For instance, choosing the Gumbel loss $f(x) = e^x-x-1$ assumes that the underlying error follows a Gumbel distribution. This will yield a weighting function of $\omega^*\propto \max(0,1-e^{-A})$. This choice is more conservative than squared loss and reduces sensitivity to outliers, but could also exacerbate the problem of mode coverage due to overly conservative updates.
>
> > Are there any empirical results on the policy learning performance with different $f$ function?
>
> We will add an experiment of the Gumbel loss function to show the robustness of the proposed framework. Here we keep the evaluation metric the same and perform parameter search over $\alpha\in [0.5,1.0,2.0]$, $\lambda\in[0.0,0.4, 0.8]$. While the squared-loss instantiation remains the strongest default in these tasks, the alternative $f$ is competitive, supporting that the framework is not tied to a single special-case loss. We will include in the camera-ready this sensitivity analysis and a short discussion of the resulting weight-shape trade-offs.
>
> |            | Hopper-m  | Halfcheetah-m | Walker2d-m | Hopper-m-r | Halfcheetah-m-r | Walker2d-m-r | Hopper-m-e | Halfcheetah-m-e | Walker2d-m-e |
> | ---------- | --------- | ------------- | ---------- | ---------- | --------------- | ------------ | ---------- | --------------- | ------------ |
> | REG-Gumbel | 77.5± 7.3 | 47.2±0.2      | 84.3±2.6   | 98.1 ±3.5  | 45.4±0.5        | 83.2±7.4     | 107.6±2.1  | 93.2±1.2        | 110.2±0.6    |
> | REG        | 91.9± 6.7 | 53.6± 0.5     | 87.9 ± 1.7 | 101.4±0.5  | 50.0±0.2        | 87.3±4.3     | 110.6±0.9  | 94.1±0.6        | 113.5±0.8    |
>
> > What is the intuition behind the policy gradient $g_{pg}$?
>
> We apologize for any confusion this may have caused. The policy gradient $g_{pg}(\theta)$ equation in Sec. 4.4 had a typo. The correct formulation is:
>
> $$g_{pg}(\theta) = E_{s\sim \mathcal{D},a \sim{\pi_\theta(\cdot|s)}}[A(s, a)\nabla_\theta\log\pi_\theta(a\mid s)]$$
>
> which corresponds to the gradient w.r.t. the training policy model of the expected cumulative rewards, as evaluated by the learned critic, that is, $\nabla_\theta E_{s\sim \mathcal{D},a\sim\pi_\theta(\cdot|s)}[Q(s,a)]$. In contrast with trust-region methods that importance-sample the expectation using the frozen current policy, this is a standard policy-gradient method that employs the log-derivative (a.k.a REINFORCE) trick.

---

> > ### Author Rebuttal · Reviewer_64Y5 · 2026-04-04
> >
> > I thank the authors for the reply and explanation. All my concerns have been adequately addressed.

---

> > > ### Author Response · Authors · 2026-04-07
> > >
> > > Thank you for your message and for confirming that our rebuttal and explanations addressed your concerns. We sincerely appreciate your time and the constructive feedback you provided during the review process, which has been instrumental in strengthening the paper.

---

### Decision · Program_Chairs · 2026-04-30

**Decision:**

Accept (regular)

**Comment:**

This work studies offline RL. Existing methods often require accurate estimation of the behavior policy, which can be challenging in practice. Instead, this work addresses this issue through a robust optimization formulation. A second contribution lies in policy extraction, where the authors propose a new OPG algorithm that explicitly encourages the learned policy to deviate from the behavior policy. Overall, this work provides a solid improvement over IQL-type methods. I agree with the reviewers that the paper does a good job of supporting its claims with both theoretical analysis and empirical results. However, given the large number of relevant offline RL baselines, the reviewers also raised a valid concern that several important comparisons are still missing from the current manuscript. Therefore, I recommend the authors to provide more experiment results in revision.